# Electronic metal-support interaction enhanced oxygen reduction activity and stability of boron carbide supported platinum

Colleen Jackson[1], Graham T. Smith[1,2], David W. Inwood[3], Andrew S. Leach[3], Penny S. Whalley[3], Mauro Callisti[2], Tomas Polcar[2], Andrea E. Russell[3], Pieter Levecque[1] & Denis Kramer[2]

Catalysing the reduction of oxygen in acidic media is a standing challenge. Although activity of platinum, the most active metal, can be substantially improved by alloying, alloy stability remains a concern. Here we report that platinum nanoparticles supported on graphite-rich boron carbide show a 50–100% increase in activity in acidic media and improved cycle stability compared to commercial carbon supported platinum nanoparticles. Transmission electron microscopy and x-ray absorption fine structure analysis confirm similar platinum nanoparticle shapes, sizes, lattice parameters, and cluster packing on both supports, while x-ray photoelectron and absorption spectroscopy demonstrate a change in electronic structure. This shows that purely electronic metal-support interactions can significantly improve oxygen reduction activity without inducing shape, alloying or strain effects and without compromising stability. Optimizing the electronic interaction between the catalyst and support is, therefore, a promising approach for advanced electrocatalysts where optimizing the catalytic nanoparticles themselves is constrained by other concerns.

[1] HySA/Catalysis, Catalysis Institute, Department of Chemical Engineering, University of Cape Town, Corner of Madiba Circle and South Lane, Rondebosch 7701, South Africa. [2] Engineering Sciences, University of Southampton, University Road, Southampton SO17 1BJ, UK. [3] Department of Chemistry, University of Southampton, University Road, Southampton SO17 1BJ, UK. Correspondence and requests for materials should be addressed to D.K. (email: d.kramer@soton.ac.uk).

The oxygen reduction reaction (ORR) is a standing grand challenge in fundamental research in energy, exemplifying a broader class of multi-electron reactions[1]. Oxygen reduction efficiency and long-term catalyst stability are standing fundamental problems that have to be overcome for polymer electrolyte fuel cells[2] to be used in demanding applications such as automobiles and grid-scale energy storage. The relatively large overpotential of the ORR at technically feasible electrodes under practical operating conditions limits conversion efficiency to typically <50% relative to the thermodynamic limit of more than 80% (ref. 3). Challenging cost targets, especially in the automotive industry, provide further motivation to reduce Pt loadings[4]. Finally, the dissolution and/or loss of Pt surface area in the cathode must be greatly reduced to meet automotive and grid-scale storage stability targets.

The work pioneered by Nørskov[5] has provided a rational design approach to improve ORR activity by tailoring the position of the Pt d-band centre, because it determines the stability of reaction intermediates, especial $O_{ads}$. Significant efforts have successfully focused on improving ORR activity by d-band centre engineering via nanoparticulate Pt alloys supported on high-surface area carbon[2,6–8]. However, the most active alloys[3,9] are unlikely to show long-term stability under cathodic polymer electrolyte fuel cells conditions[10,11] for thermodynamic reasons. At best, one can hope that dealloying is restricted to the surface, leading to Pt skin structures that are, fortuitously, highly active[3,12,13].

Whether dealloying can be restricted to the surface of the typically 2 nm alloy electrocatalysts over technically relevant lifetimes is an open question. Regardless, stability concerns motivate other design approaches to engineer the Pt d-band characteristics without potentially compromising stability. Recently, research emphasis has shifted to alternative support materials to prevent the loss of catalyst surface area due to support corrosion and dissolution/agglomeration of nanoparticulate Pt (refs 14,15). Alternative supports also provide an opportunity to indirectly improve ORR activity as well, because a change in support electronic structure can influence the catalyst's d-band manifold via electronic equilibration[16], one aspect of strong metal-support interactions (SMSI).

Utilization of SMSIs is a promising approach to improve both activity and stability of supported Pt catalysts. The SMSI of Pt on oxide supports, particularly titanium dioxide, is well known[17,18]. These $Pt/TiO_2$ catalysts have previously exhibited higher activities for ORR in relation to Pt/C (ref. 19). The low conductivity of these materials, however, is a challenge[20]. Highly conductive $SnO_2$-based supports have, therefore, attracted attention recently[21], and thin film studies unveiled an intriguing dependence of ORR activity on support orientation and/or termination[22].

Changing the physical nature of the support from carbons to oxides is a radical step. It potentially impacts catalyst stoichiometry and particle morphology, which in their own right change electrocatalytic properties via size[4], strain[23] and proximity[24] effects. It is, therefore, nontrivial to experimentally isolate these effects to develop the fundamental understanding needed to rationally exploit SMSIs.

Transition metal carbides have also been widely investigated as promising support materials due to high electrical conductivity, thermal stability and similarities in electronic structure to Pt (ref. 25). However, the low surface area of carbides and liability to oxidation are challenging[26]. Carbides are, nonetheless, attractive to investigate electronic metal-support interactions in isolation, because they are physically closer to carbon than oxides. The material of choice, however, should also show a markedly different electronic structure and resilience against oxidation. Boron carbide ($B_4C$) is such a material. It is a refractory compound[26] exhibiting high chemical inertness[27], semi-conducing band structure and electrochemical stability[28]. The first study of $B_4C$ as an electrocatalyst support for hydrogen fuel cells was by Grubb & McKee[29] wherein a high resistance to sintering and higher current densities when compared to platinum on carbon black were already observed. More recent work by Lv et al.[30] has also shown promising ORR activity.

Pt nanoparticle supported on graphite-rich $B_4C$ composites (BC) are compared here with Pt supported on Vulcan XC 72R high-surface area carbon (C) to isolate electronic metal-support interactions. Electrochemical as well as advanced ex situ characterization techniques are used to elucidate the structure-property relationships between catalyst morphology, electronic structure and ORR activity. Transmission electron microscopy (TEM) and extended X-ray absorption fine structure (EXAFS) analysis confirm similar particle size, shape and packing of the clustered nanoparticles on both supports, while X-ray photoelectron spectroscopy (XPS) and X-ray adsorption near edge structure (XANES) analysis show differences in electronic properties due to metal-support interactions. These differences in electronic properties are correlated with enhanced ORR activity: the surface and mass specific oxygen reduction activity of Pt nanoparticles supported on BC under rotating disc electrode (RDE) conditions increases by roughly 40% relative to commercial Pt/C, while the kinetic current at 0.9 V shows a 50–100% improvement. The increased cycle stability of the BC supported catalysts after degradation cycling further promotes the interpretation in terms of increased electronic metal-support interactions on these catalysts. Purely electronic metal-support interactions are, therefore, experimentally isolated demonstrating that optimizing the electronic interaction between the catalyst and support, in conjunction with d-band theory, is a promising approach to designing advanced electrocatalysts without compromising stability under harsh conditions.

## Results

**Naming scheme.** Graphite-rich boron carbide composites (BC) were catalysed with varying amounts of Pt nanoparticles following standard procedures. These Pt/BC catalysts are compared with commercial Pt/C catalysts. It is imperative to keep particle size and dispersion constant between the Pt/BC and benchmark Pt/C catalysts to isolate electronic effects, because these affect ORR activities significantly[4,23–24]. We use a naming convention for all catalysts that follows the nominal platinum loading (that is, wt%) of the commercial benchmark catalysts. The equivalent platinum loading on BC (that is, eq. wt%) designates a catalyst that has the same mass platinum per support surface area (Supplementary Note 1).

**Support surface area.** The BC support Brunauer, Emmett and Teller (BET) surface area was measured to be 80 $m^2 g^{-1}$, in agreement with manufacturer specifications (Supplementary Fig. 1). This is considerably lower than seen for high-surface area carbons such as Vulcan XC 72R, which has a surface area of 250–260 $m^2 g^{-1}$. To keep the mass platinum per surface area constant, the actual target platinum loading per mass BC support was, therefore, reduced accordingly (Supplementary Table 1) to maintain a similar Pt coverage per surface area of support and particle size as seen on the commercial Pt/C benchmark catalysts to isolate support-related activity changes.

**X-ray diffraction.** The X-ray diffraction pattern of BC (Supplementary Fig. 2) clearly shows peaks indicative of rhombohedral graphite (3R) and hexagonal graphite (2H) in

addition to rhombohedral $B_4C$ ($R\bar{3}m$), showing the material to be a composite of graphite and rhombohedral $B_4C$. Using Rietveld refinement, the spectrum indicates $58.0 \pm 7.7$ wt% graphite, $38.7 \pm 7.2$ wt% $B_4C$ and $3.0 \pm 1.7$ wt% $B_2O$ in the sample. This is in reasonable agreement with the average B:C ratio of 39:61 as obtained from EELS. The high graphite content ensures comparative conductivities between BC and C (Supplementary Note 2). The graphite consists of approximately 32.0 wt% rhombohedral graphite (3R) and 26.0 wt% hexagonal graphite (2H).

The average Pt crystallite size was confirmed by a fit of the Scherrer equation to the Pt(200), Pt(220) and Pt(311) peaks at $46°$, $67°$ and $81°$ in the X-ray diffraction (XRD) spectra, respectively (Supplementary Fig. 3). This analysis yielded a volume-weighted average particle size of $\sim 3.9$ nm for the various loadings, in good agreement with the size measured via TEM (Table 1). XRD routinely overestimates particle size compared to TEM at the relevant length scale, because particle sizes smaller than 2 nm are not identified due to the lower detection limit of XRD (ref. 31).

**Transmission electron microscopy.** TEM images of the prepared Pt/BC catalysts as well as the as-received commercial Pt/C benchmark catalysts, taken at 270 k times magnification to illustrate particle dispersion and support morphology, are shown in Fig. 1. The Pt loadings on BC were confirmed via ICP-OES analysis to be 10, 20 and 40 eq. wt%, respectively (see Supplementary Table 1 for actual loadings). Pt particle size distribution and dispersion for each catalyst are quantified and contrasted in Table 1, which shows average particle size with s.d. as well as average interparticle distances with s.d.

The BC support particles are homogenous with an average particle size of $\sim 40$ nm, and Pt nanoparticles are well dispersed on the BC support with average particle sizes ranging from 2.3 to 2.9 nm. Particle size and interparticle distances are very similar for equivalent loadings on both supports at 20 wt% and 40 wt% with little s.d. Non-conformity is seen between the 10 wt% Pt/C and 10 eq wt% Pt/BC catalysts due to the unusually small Pt size on Pt/C.

**X-ray absorption spectroscopy.** The EXAFS was collected for the 20 eq. wt% Pt/BC and 20 wt% Pt/C catalysts. The EXAFS at the Pt $L_3$ edge gives insights to the supported Pt morphology: the first shell co-ordination number is strongly related to average particle size[32] while the fourth co-ordination number is an indication of particle shape.

The measured and modelled EXAFS spectra are shown in Supplementary Fig. 4. The fitted model, considering up to the fourth neighbour shells, describes the EXAFS data well. The resulting co-ordination numbers ($N$), distances of neighbouring atoms ($R$) and the disorder factors ($\sigma^2$) of the catalysts supported on BC and C are listed in Table 2.

The first, second and fourth Pt co-ordination numbers are slightly smaller for Pt supported on BC, while the third is found to be slightly larger. Differences, however, are within experimental error indicating very similar Pt morphologies as further discussed later. Near-neighbour distances also agree very well between the Pt/BC and Pt/C catalysts.

**X-ray photoelectron spectroscopy.** Figure 2a shows the C 1s and Pt 4f regions of XPS spectra for the Pt/BC and Pt/C catalysts. The XPS spectra are calibrated via alignment of the $C(sp^2)$ peak position in the C 1s spectrum to its reference value of 284 eV.

The C 1s region of the Pt/C catalyst is dominated by the $C(sp^2)$ peak with a small, broad feature at higher binding energies that

**Table 1 | TEM average particle size and proximity.**

| Catalyst | TEM average particle size (nm) | Average interparticle distances (nm) |
|---|---|---|
| 10 eq wt% Pt/BC | $2.9 \pm 0.9$ | $18.6 \pm 11$ |
| 20 eq wt% Pt/BC | $2.3 \pm 0.7$ | $7.94 \pm 3.6$ |
| 40 eq wt% Pt/BC | $2.6 \pm 0.6$ | $5.35 \pm 1.6$ |
| 10 wt% Pt/C | $1.1 \pm 0.3$ | $6.55 \pm 2.8$ |
| 20 wt% Pt/C | $2.1 \pm 0.5$ | $7.09 \pm 2.1$ |
| 40 wt% Pt/C | $2.7 \pm 0.7$ | $5.09 \pm 1.3$ |

Particle size and interparticle distance distribution mean and s.d. calculated from TEM images.

has been assigned to C–O bonding structures of carboxyl groups[33]. The C 1s spectrum of the Pt/BC catalysts is a convolution of contributions from the $B_4C$ and graphitic carbon within the composite. The $C(sp^2)$ peak is assigned to the graphitic carbon, and the peaks at 282.7 and 284.8 eV are attributed to $B_4C$. The low energy polymorphs of $B_4C$ (ref. 34) provide three C environments at the ends and centre of tri-atomic chains and in the polar position within $B_{11}C$ icosahedra. The low binding energy peak at 282.7 eV was previously attributed to $B_4C$ (ref. 35) and is likely due to carbon in low co-ordination (chain centre and/or $B_{11}C$). The $C(sp^3)$ peak at 284.8 eV, which is reminiscent of diamond[36], is also predominantly assigned to $B_4C$ due to the strong similarity of site environment of tetrahedral carbon at the chain ends in $B_4C$ and diamond and the lack of evidence for $sp^3$ hybridized carbon from XRD.

This assignment is supported by density functional theory calculations (Supplementary Note 3) that predict a chemical shift of $\sim 2.3$ eV between the C 1s core electrons in $B_{11}C$ and CBC units (Supplementary Table 2), in good agreement with the experimental shift of 2.1 eV. Hence, the relative strength of the $C(sp^2)$ and $C(sp^3)$ peaks corroborates the phase fractions of $B_4C$ and graphitic carbon within the composite derived from XRD.

The Pt 4f spectra of the Pt/C catalyst features the usual spin–orbit splitting into a $4f^{7/2}$ peak at 71.0 eV and a $4f^{5/2}$ peak at 74.3 eV. The same splitting is seen for all Pt/BC catalysts. There is, however, a clear shift of both peaks by $\sim 0.6$ eV to higher binding energies for all Pt loadings on BC to 71.6 and 74.9 eV.

***In situ* x-ray absorption near edge structure.** Figure 2c shows the XANES Pt $L_2$ and $L_3$ edges for the 20 wt% Pt/C and 20 eq. wt% Pt/BC catalysts measured under potentiostatic control in electrochemical environment. The increase in the $L_2$ white line observed for the Pt/BC catalyst indicates an increase in unoccupied d-states above the Fermi level for Pt/BC (ref. 37) in electrochemical environments. Using standard post-processing[38], d-band occupancy can be estimated to be 63% for Pt/C and 58% for Pt/BC at 0.744 V versus NHE.

**Electrochemical characteristics and activity.** The calculated electrochemically active surface area (ECSA) and ORR activities are summarized in Table 3. CO stripping voltammetry (Supplementary Fig. 5) was used to determine the ECSAs assuming a specific charge of $420\ \mu C\,cm_{Pt}^{-2}$ per CO monolayer adsorbed on the platinum surface. Measurements were taken at room temperature in 0.1 M $HClO_4$ electrolyte solution. No CO stripping peak was seen on the uncatalysed BC support. ECSAs were confirmed using Cu-UPD[39].

The specific and kinetic currents reported in Table 3 were obtained from RDE measurements at a rotation speed of 1,600 r.p.m. (Supplementary Fig. 6) during the anodic sweep at 0.9 V versus RHE. Measurements were completed at 20 mVs$^{-1}$ to minimize double layer charging artefacts at room temperature

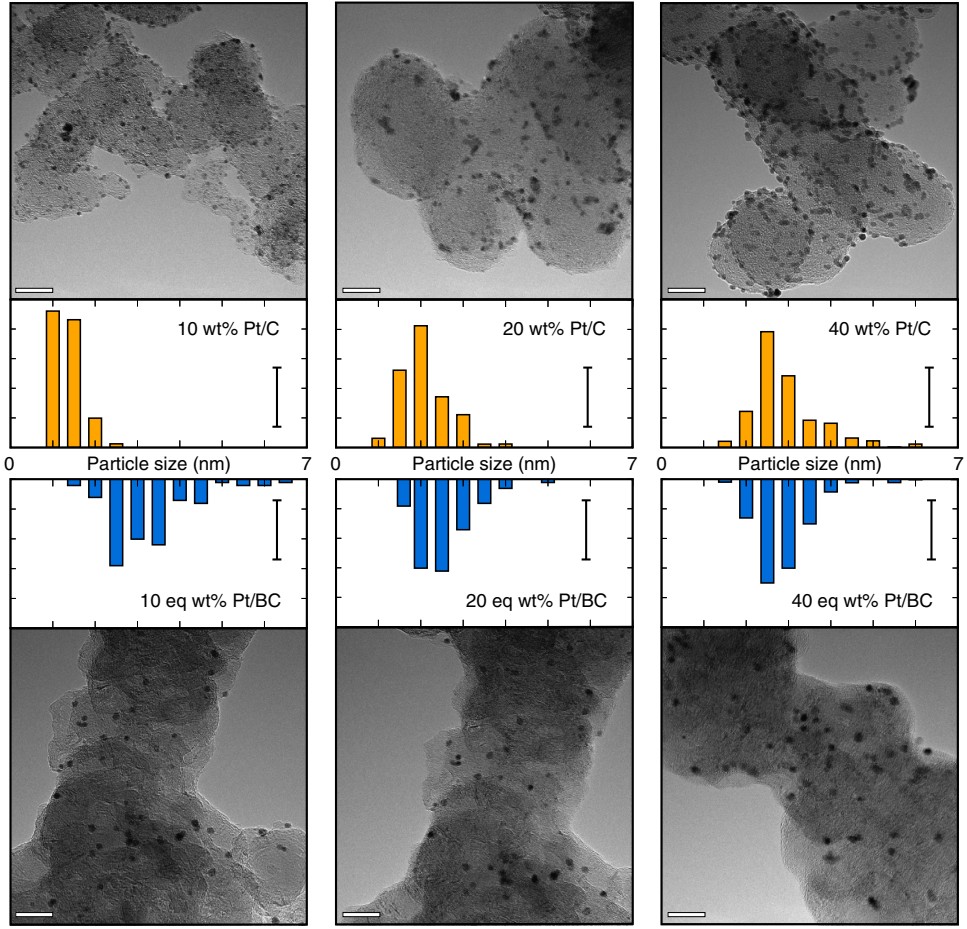

**Figure 1 | Catalyst particle size and dispersion.** TEM images taken at 270 k magnification showing Pt particle size and dispersion on the C and BC supports, TEM scale bars represent 20 nm; histogram scale bars represent 20%; histograms show Pt size distribution obtained from 100 particles across three images each.

**Table 2 | Co-ordination numbers and distances of neighbouring atoms.**

| Catalyst | Neighbouring atom | $N$ | $R$ (Å)* | $\sigma^2/10^3$ |
|---|---|---|---|---|
| 20 eq wt% Pt/BC | Pt—Pt$_1$ | 9.9 ± 0.3 | 2.751 | 5.8 ± 0.15 |
| | Pt—Pt$_2$ | 4.2 ± 1.1 | 3.891 | 8.4 ± 1.7 |
| | Pt—Pt$_3$ | 12.8 ± 3.3 | 4.766 | 8.2 ± 1.2 |
| | Pt—Pt$_4$ | 8.0 ± 1.2 | 5.502 | 9.8 ± 0.78 |
| 20 wt% Pt/C | Pt—Pt$_1$ | 10.1 ± 0.3 | 2.749 | 5.9 ± 0.17 |
| | Pt—Pt$_2$ | 5.1 ± 1.4 | 3.888 | 10 ± 2.1 |
| | Pt—Pt$_3$ | 11.2 ± 3.3 | 4.762 | 8.1 ± 1.4 |
| | Pt—Pt$_4$ | 8.3 ± 1.3 | 5.498 | 10.5 ± 0.95 |

*The error associated with distance to neighbouring atoms for the 20 eq wt% Pt/BC and 20 wt% Pt/C is ± 0.00048 Å and ± 0.00056 Å, respectively.
Values and confidence intervals obtained from EXAFS fits shown in Supplementary Fig. 4, disorder factors $\sigma^2$ are given in the fifth column.

in an oxygen saturated 0.1 M HClO$_4$ electrolyte[4,40]. The ECSA obtained from CO stripping voltammetry was used to calculate surface area specific currents, and the kinetic current was isolated from mass transport limitations using standard post-processing[41].

The Pt/C catalyst activities fall in line with numerous reported values and compilations of values for ORR activity on these catalysts[4,42–45]. An exception is seen for the 10 wt% Pt/C catalyst, which exhibited large experimental scatter and anomalously small limiting currents during RDE measurements (Supplementary Fig. 6a) indicative of various detrimental effects[43]. The 10% Pt/C catalyst is, therefore, not considered a suitable benchmark for our purposes.

The ORR activities for the BC supported catalysts consistently show a more active catalyst with higher mass specific ($I_{mass}$), surface area specific ($I_{spec}$) and kinetic activities ($I_k$) for the ORR. Additionally, kinetic activity increases as Pt loading increases on both the Pt/C and Pt/BC catalysts, with the least active ORR catalyst being 10 wt% Pt/C and the most active being 40 eq wt% Pt/BC. These trends are further discussed below.

Further electrochemical characteristics are summarized in Fig. 3. Cyclic voltammograms of the BC supported catalysts are shown in the left diagram of Fig. 3a. In addition to the typical characteristics of supported Pt, a slight coupled reversible reaction at ~0.6 V versus RHE is seen for the Pt/BC materials.

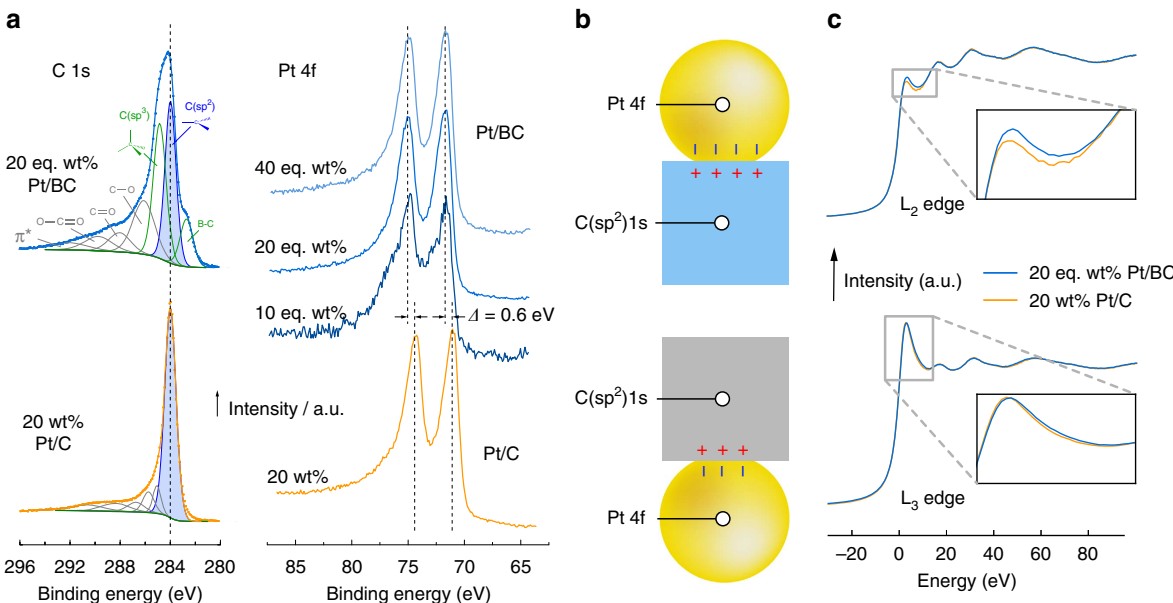

**Figure 2 | Electronic equilibration across the catalyst support interface.** (**a**) XPS spectra of the C 1s and Pt 4f region of the Pt/C and Pt/BC catalysts, spectra are aligned to the graphitic C (sp$^2$) peak at 284 eV; (**b**) Schematic of charge transfer across the support-catalyst interface due to Fermi level equilibration rationalizing relative shifts in the XPS Pt 4f and C 1s core levels; (**c**) *In situ* XANES L$_2$ and L$_3$ edge measured at 744 mV versus NHE probing d-band occupancy in electrochemical environment.

**Table 3 | Calculated ECSA and ORR activity.**

| Catalyst | ECSA (m$^2$ g$_{Pt}^{-1}$) | $I_{mass\ (0.9\,V)}$ (Ag$_{Pt}^{-1}$) | $I_{spec\ (0.9\,V)}$ (Am$_{Pt}^{-2}$) | $I_k$ (Am$_{Pt}^{-2}$) |
|---|---|---|---|---|
| 10 eq wt% Pt/BC | 81.1 ± 3.9 | 224 ± 21 | 2.8 ± 0.1 | 3.2 ± 0.3 |
| 20 eq wt% Pt/BC | 67.7 ± 1.9 | 172 ± 7.0 | 2.6 ± 0.2 | 3.7 ± 0.3 |
| 40 eq wt% Pt/BC | 69.5 ± 0.4 | 209 ± 4.0 | 3.0 ± 0.1 | 7.1 ± 0.3 |
| 10 wt% Pt/C | 115 ± 15 | 146 ± 66 | 1.2 ± 0.5 | 1.5 ± 0.7 |
| 20 wt% Pt/C | 82.8 ± 2.5 | 156 ± 4.5 | 1.9 ± 0.1 | 2.4 ± 0.1 |
| 40 wt% Pt/C | 56.0 ± 3.6 | 120 ± 3.5 | 2.2 ± 0.1 | 3.3 ± 0.2 |

ECSA determined by CO stripping and ORR activity measured at 0.9 V versus RHE in a 0.1 M HClO$_4$ electrolyte solution at room temperature. Error margins (s.d.) were obtained from 2 to 4 repeats for each data point.

The large double layer of the Pt/BC catalysts results in a less well defined H$_{UPD}$ region than seen on the Pt/C catalysts. Therefore, no attempt was made to compare the ECSA as obtainable from the H$_{UPD}$ region with the values given in Table 3. Careful observation of the cyclic votammograms in Fig. 3b, which compares the electrochemical response of Pt/BC with the Pt/C benchmark at 40 wt%, shows slight peak shifts towards higher potentials in the oxide reduction region for Pt/C.

The kinetic ORR results are reiterated in the Tafel plots shown to the right in Fig. 3 for completeness. Figure 3a compares kinetic currents of Pt/BC at various loadings. Again, kinetic activity generally increases with Pt loading over the full potential range although the two lower loadings show comparable kinetic currents at lower voltages. Finally, the Tafel plot of Fig. 3b shows that the 40 eq wt% Pt/BC catalyst exhibits higher kinetic activity for ORR over the measured potential range of 0.85–0.95 V versus RHE than the 40 wt% Pt/C benchmark. In fact, relative activity of Pt/BC versus Pt/C seems to increase further with decreasing potential.

**Cycle stability.** Catalyst durability was studied for the most active Pt/BC and Pt/C catalysts to investigate the platinum dissolution and platinum agglomeration rate on the different supports. Durability testing consisted of cycling the potential between 0.6 and 1.0 V versus RHE for 6,000 cycles at a scan rate of 50 mVs$^{-1}$ and periodically returning to the full voltammogram scan (0.05–1.2 V versus RHE at 20 mVs$^{-1}$) to compare changes in the H$_{UPD}$ region and oxygen adsorption and desorption region.

The voltammograms of the Pt/BC catalyst show significantly less change after 6,000 cycles compared to Pt/C (Fig. 4a). Changes in the 1.0–1.2 V range for Pt/BC appear to be not purely related to ECSA loss, because less variation is seen in the H$_{UPD}$ region after cycling. Instead, the reduced current response in the oxidative region is in parts attributed to secondary conditioning effects, which could either be the removal of organic impurities from Pt or oxidative decomposition of other trace impurities in solution amongst other things.

Cyclic voltammograms of Pt/C before and after cycling show a significantly altered current response in the oxide as well as in the H$_{UPD}$ region while the double layer region remains invariant after 6,000 cycles, strongly suggesting catalyst degradation and ECSA loss.

The ECSAs shown in Fig. 4b were periodically obtained from the H$_{UPD}$ region. Both catalysts show a relatively stable ECSA for an initial period of ∼50–100 cycles, which might be due to cleaning and/or roughening effects offsetting initial degradation. A logarithmic loss of ECSA is seen for both catalysts beyond this initial conditioning phase. Focusing on the data from cycle 300

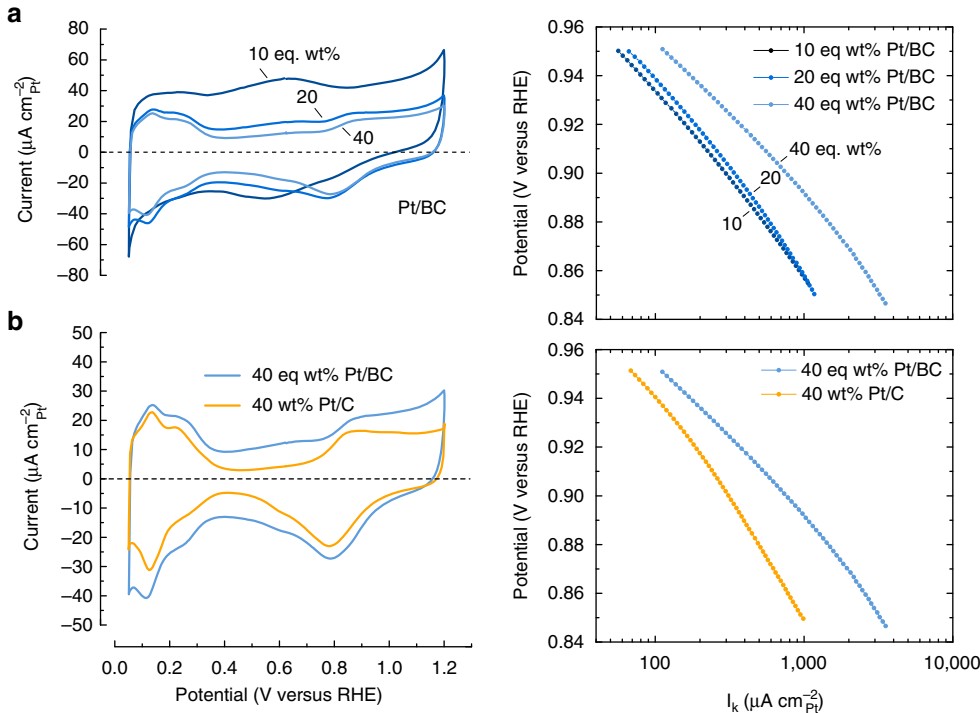

**Figure 3 | Electrochemical characterization.** (**a**) Shows cyclic voltammograms and Tafel plots for 10, 20 and 40 eq wt% Pt/BC, and (**b**) compares 40 eq wt% Pt/BC with 40 wt% Pt; CVs recorded at 20 mVs$^{-1}$ between 0.05 and 1.2 V versus RHE and normalized for surface area specific current, Tafel plots obtained from anodic scan recorded at 1,600 r.p.m. between 0.05 and 1.2 V versus RHE and normalized for surface area specific current; all experiments performed in 0.1 M HClO$_4$ at room temperature.

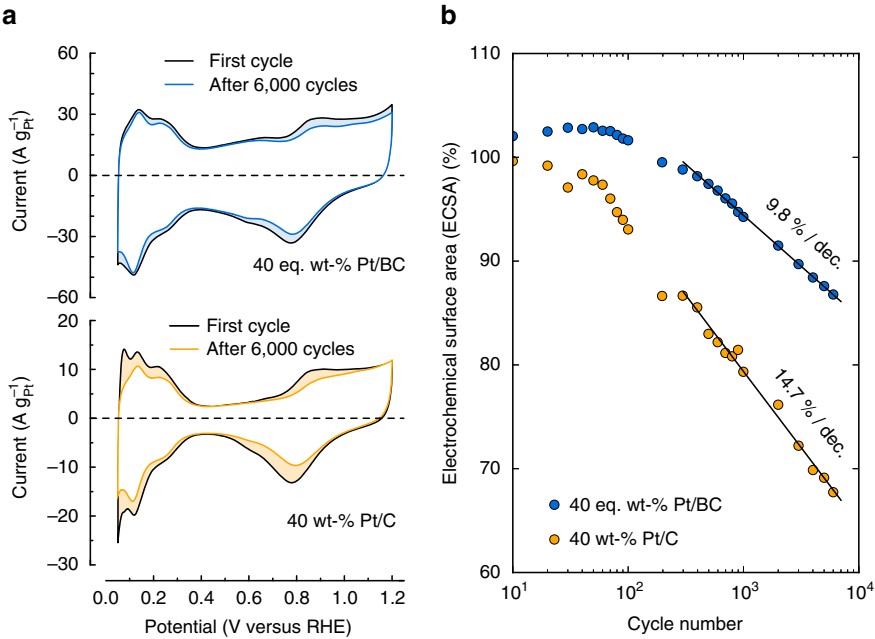

**Figure 4 | Catalyst cycle stability.** (**a**) Compares CVs between the first and 6,000th cycle, and (**b**) plots the change in ECSA after cycling at 50 mVs$^{-1}$ between 0.6 and 1.0 V versus RHE in 0.1 M HClO$_4$ at room temperature.

onwards, the Pt/C catalyst loses 14.7% ECSA per decade, while the Pt/BC catalysts shows only 9.8% ECSA loss per decade, a 1/3 reduction in logarithmic degradation rate.

ORR activity of cycled catalysts as obtained from RDE measurements (Supplementary Fig. 7) is compared with ORR activity of the pristine catalysts in Table 4 for one set of catalysts. For consistency, specific and kinetic activities are based on ECSAs

measured again after 6,000 cycles by CO stripping (Supplementary Fig. 8). The ECSAs reported in Table 4 agree well with the cyclic H-UPD measurements (Fig. 4), showing a significantly larger loss in surface area for the Pt/C catalyst (23% loss after 6,000 cycles) than for the Pt/BC catalyst (12% loss after 6,000 cycles). Accordingly, mass activity of the cycled Pt/BC catalyst remains with 168A per g$_{Pt}$ ~30% higher than seen for

**Table 4 | Catalyst ORR activity loss.**

| Catalyst | ECSA ($m^2 g_{Pt}^{-1}$) | $I_{mass\,(0.9\,V)}$ ($Ag_{Pt}^{-1}$) | $I_{spec\,(0.9\,V)}$ ($Am_{Pt}^{-2}$) | $I_k$ ($Am_{Pt}^{-2}$) |
|---|---|---|---|---|
| *40 eq wt% Pt/BC* | | | | |
| Before cycling | 69.8 | 199 | 2.89 | 6.95 |
| After cycling | 61.3 | 168 | 2.74 | 5.43 |
| Relative change (%) | −12 | −15 | −5 | −22 |
| *40 wt% Pt/C* | | | | |
| Before cycling | 54.4 | 139 | 2.55 | 4.24 |
| After cycling | 41.8 | 126 | 3.01 | 4.71 |
| Relative change (%) | −23 | −9 | +18 | +11 |

ECSA determined by CO stripping and ORR activity measured at 0.9 V versus RHE in a 0.1 M $HClO_4$ electrolyte solution at room temperature before and after 6,000 cycles; mass specific values refer to initial Pt loading.

**Figure 5 | Catalyst particle shape.** Co-ordination numbers $N_1$ through $N_4$ for icosahedra and cuboctahedra with superimposed calculated co-ordination numbers from the EXAFS for the 20 eq wt% Pt/BC and 20 wt% Pt/C against average particle size as suggested by TEM. Error bars indicate s.d. of TEM particle size distribution and confidence intervals from EXAFS fit. Shape models adapted from Glasner and Frenkel[48].

the cycled Pt/C catalyst. Relative loss of mass, specific and kinetic activity of the Pt/BC catalyst are in reasonable agreement with the loss in ECSA for the Pt/BC catalyst. The Pt/C catalyst, however, shows a markedly different behaviour. While mass activity also degrades, the relative loss in mass activity (9% loss) is smaller than the reduction in ECSA (23% loss). Correspondingly, surface specific and kinetic activities increase for reasons discussed later. Nonetheless, kinetic activity of the cycled Pt/C catalyst remains significantly lower than seen for the Pt/BC catalyst after 6,000 cycles.

**Platinum nanoparticle size and shape.** Platinum was successfully deposited onto BC with the same average particle sizes as on carbon supports (with the exception of 10 wt% Pt/C) and similar particle size distributions (Table 1 and Fig. 1). There is no significant increase in particle size for the higher loadings, indicating that the surface area for both supports is sufficient to accommodate high loadings of nanoparticles.

Figure 5 compares the Pt–Pt co-ordination numbers obtained from EXAFS results presented in Table 2 with co-ordination numbers for ideal icosahedra and cubocahedra[46]. The first four Pt–Pt co-ordination numbers are plotted against average number of atoms per particle as obtained from TEM. The error margins are calculated from the TEM particle size distributions and the confidence interval of the modelled parameters as given by Demeter[47].

The co-ordination numbers $N_{1...4}$ for the two supports are within the error margins of the fits and, therefore, suggest very similar Pt particle sizes and shapes on both supports. $N_1$ is often used to deduce particle sizes[40]. Comparing $N_1$ with the expectation values for cuboctahedra and icosahedra shows good agreement with the TEM derived particle sizes for both catalysts.

Particle shape has a stronger influence on more distant platinum shells[32]. $N_2$ and $N_3$, however, do not lend themselves to further analysis in our case. The differences between icosahedral and cuboctahedral geometries are too small relative to the precision of our fitted values, which is in part inherently limited by the particle size distribution. For this reason, we have abstained from using the method described by Jentys[48], which relies on the ratio of $N_3$ to $N_1$ to deduce particle size and shape. The analysis of $N_4$, however, is instructive due to the strong shape sensitivity of $N_4$ for icosahedra and cuboctahedra. The $N_4$ values of both catalysts agree better with a cuboctahedal shape model, suggesting similar exposure of Pt(001) and Pt(111) facets on both supports. Consequently, EXAFS suggests even stronger than TEM that sample averaged particle sizes and shapes are very similar for the Pt/BC and Pt/C catalysts.

**Electronic metal-support interaction.** As there is no significant change in catalyst size, shape and lattice constant, it is tempting to attribute the significant changes in ORR activity to electronic effects. The Pt 4f spectrum shifts to higher binding energies

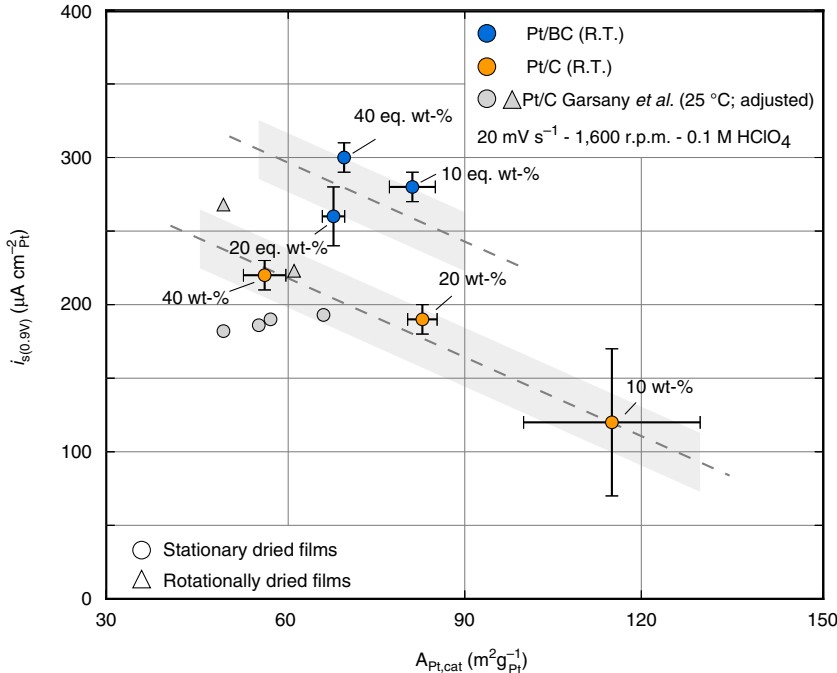

**Figure 6 | Comparison of surface area specific ORR activities.** Specific current densities at 0.9 V versus RHE for the Pt/BC and Pt/C catalysts are compared with reference data for ORR activity of similar Pt/C catalysts from Garsany et al.[44,45] adjusted for temperature[60] and $O_2$ mass transport correction.

relative to the C 1 s core levels for Pt/BC (Fig. 2a). Following Lewara et al.[17], this might indicate a possible alloy formation with boron or a change in charge transfer between Pt and the support material. There is no evidence of alloy formation in the B 1s XPS (Supplementary Fig. 9). The B 1s peak does not shift, and a model considering only B–B, B–C and B–O bonds is in perfect agreement with the observed spectra. The very similar Pt lattice constants obtained from EXAFS (Table 2) provide further evidence for pure Pt particles. The Pt 4f binding energy shift is, therefore, assigned to purely electronic effects.

It is interesting to note that similar electronic effects have been observed for multiwall carbon nanotubes by Ma et al.[49] who find a similar but smaller shift of the Pt 4f core levels. These carbons typically show a work function ~0.3–0.5 eV different from graphite[50]. It is, therefore, likely that the measured shift in the Pt 4f core levels are a direct result of changes in the support electronic structure.

Generally, charge is transferred to equilibrate the respective Fermi levels when support and Pt nanoparticle contact. This is qualitatively shown in Fig. 2b. Amongst other factors, the magnitude of the expected charge transfer will depend on the electronic properties of the support material. When changing the support (in this case from C to BC) the extent of this charge transfer changes, resulting in changes in the valence manifolds of the supported catalyst[51] and/or support[52]. This change in electron occupancy can be probed using relative shifts of core levels on either side of the interface[52].

By aligning the C($sp^2$) 1s levels for Pt/C and Pt/BC to exclude other effects, a shift in the Pt 4f peak reflects changes in valence state occupancy across the interface as a result of the changing support material, but might also be influenced to some degree by varying final state effects[52]. The reduction of the difference between Pt 4f and C($sp^2$) 1s core levels by ~0.6 eV shown in Fig. 2, therefore, indicates an increase of Pt 4f and/or decrease of C 1s core level binding energies due to larger electron transfer into Pt under UHV conditions, strongly suggesting that the

improved ORR activity and stability originates from electronic metal-support interactions.

**Charge transfer and d-band occupancy.** In principle, one should be able to correlate the sign and magnitude of the relative core level shift with the electronic structure of the respective support materials. There is, however, considerable uncertainty around the electronic properties of the materials reported in the literature. For example, the work function of carbons is usually around 5 eV, but varies as a function of surface termination[53]. Further, the band structure of boron carbides is sensitive to the local stoichiometry of samples, and the Fermi level depends on the type/degree of doping. The situation is further complicated by the fact that the support material is a composite containing a significant phase fraction of graphitic carbon although we expect the phase fraction on the surface to be significantly smaller than for the bulk support materials due to the aggressive pre-treatment in nitric acid.

Regardless, stronger electron transfer should occur from $B_4C$ (typical work function ~4–4.5 eV (ref. 54)) to Pt (typical work function ~5.5 eV (ref. 55)) than from C (typical work function ~5 eV (ref. 53)) to Pt to equilibrate the respective Fermi levels as illustrated in Fig. 2b and in agreement with the metal-support interaction suggested by XPS under UHV conditions.

The situation in electrochemical environment, however, is more complex. The *in situ* XANES spectra shown in Fig. 2c clearly shows an increased hole density in the Pt d-band manifold when particles are supported on BC (that is, more positively charged particles). This apparent contradiction can be resolved by realising that *in situ* XANES probes a charged interface under potentiostatic control, while *ex situ* XPS necessarily probes the catalyst under overall neutrality. Because work function and potential of zero charge (pzc) are proportional to each other[56], the smaller support work function of the BC-based catalysts will cause a negative shift of the pzc and, hence, a more positively

charged Pt/BC catalyst if held at the same potential as the Pt/C catalyst in the same aqueous electrolyte. Finally, it is noteworthy that the support-induced decrease in d-band filling seen for Pt/BC is reminiscent of electronic effects seen for highly active Pt-alloys[57].

**Comparison of ORR activity with benchmark data.** The trends in ORR kinetic activities (Fig. 3, Table 3) are not only a function of support material. Particle size effects are likewise relevant. The conventional interpretation[58] in terms of changing surface fractions of Pt(100) and Pt(111), which show different activity for ORR[59], has recently been challenged over the relevant particle size range[40]. However, they are known to significantly impact the results from RDE measurements if the standard protocols and data treatment used here are followed.

Figure 6, therefore, compares surface area specific activity of the Pt/C and Pt/BC catalysts with benchmark data from Garsany et al.[44,45] Note that the data from Garsany et al. was corrected for temperature from 30 to 25 °C following a suggestion by Neyerlin et al.[60] and to account for the different definition of specific activity used by Garsany et al. (Supplementary Table 3). Our results for the benchmark Pt/C catalysts are in good agreement with the literature data attesting to the quality of our films and robustness of experimental procedures. As already mentioned, the 10 wt% Pt/C catalyst does not lend itself as a benchmark for our purposes due to the significantly smaller particle size and considerable experimental scatter.

The Pt/BC catalysts broadly follow a similar particle size dependence but with overall ∼50% higher surface area specific activity at 0.9 V versus RHE. The increase of surface area specific current density is significantly larger than the expected experimental scatter indicated as grey areas. We, therefore, conclude that purely electronic equilibration between support and catalyst has a bearing on ORR activity of Pt nanoparticles. This is particularly interesting if one considers that the effective charge screening of metals should completely screen the charge redistribution due to Fermi level equilibration between catalyst and support a few atomic distances away from the contact interface[61–63]. Hence, one would not directly expect a pronounced electronic effect on large parts of the more than 20 Å thick Pt particles.

**Durability.** Figure 4 shows the degradation of ECSA over 6,000 cycles for the 40 eq. wt% Pt/BC and the 40 wt% Pt/C catalyst. The voltage range was selected to promote platinum dissolution and/or agglomeration over support corrosion[64]. There is a significantly larger rate of ECSA loss for Pt/C after some initial cycles. The consequences of the logarithmic dependence seen in Fig. 4 are worth highlighting. Due to the 1/3 reduction in loss rate per decade, the Pt/BC catalyst retains more than 90% ECSA for about 2,000 cycles, while the Pt/C catalyst shows a 10% loss in surface area already after about 200 cycles.

Since particle sizes and shapes of both catalysts are initially similar, this cannot be due to a more stable particle size as starting point. Hence, the results suggest that the stronger electronic metal-support interaction between Pt and the BC support leads to a reduction in platinum dissolution and/or agglomeration. The ORR activities of the cycled catalysts reported in Table 4 suggest that mostly Pt agglomeration is inhibited for Pt/BC relative to the Pt/C catalyst. While the relative loss of mass specific ORR activity after 6,000 cycles is in line with the loss of ECSA for Pt/BC, Pt/C shows an increase in surface specific and kinetic activity after cycling. This increase in surface specific activity and simultaneous loss in mass activity is a clear indication of increasing Pt particle sizes on the Pt/C support. CO stripping voltammetry (Supplementary Fig. 8) provides further evidence for increased

Pt agglomeration on the cycled Pt/C. The pre-peak that is observed for both pristine catalysts has been attributed previously to Pt agglomerates[65]. This pre-peak feature increases for the cycled Pt/C catalyst while it disappears for the cycled Pt/BC catalyst, indicating increased particle agglomeration for cycled Pt/C.

We have argued above that a stronger dipole field is established at the Pt/BC interface relative to Pt/C, which is corroborated by XPS. This stronger dipole interaction likely reduces the mobility of Pt nanoparticles mitigating agglomeration. A similar effect is seen on carbon supports where the rate of platinum nanoparticle agglomeration is seen to reduce when carbon supports are modified with nitrogen and oxygen containing functional groups that strongly interact with nanoparticles[66,67].

Likewise, it suggests that Pt nanoparticles can adopt a lower energy state on BC, which, in principle, should mitigate dissolution. The CVs in Fig. 3b support this interpretation. The oxidation/reduction features of Pt/BC are slightly shifted to more positive potentials relative to Pt/C. Hence, the catalyst might be slightly less prone to surface oxidation/reduction and associated anodic and cathodic Pt dissolution with beneficial implications for cycle stability in this potential range.

## Methods

**Catalyst preparation.** As received $B_4C$ (NaBond Technologies Co., Ltd.) was stirred in conc. nitric acid (Kimix Chemicals and Lab Supplies cc.) at room temperature for 8 h. Organo-metallic chemical deposition was used to deposit platinum onto the treated $B_4C$ to prepare 10, 20 and 40 eq. wt% Pt/BC, where the eq. wt % was calculated from surface areas calculated from nitrogen physisorption using a TriStar II 3,020 (Micrometrics) and interpreted using BET theory. The catalyst deposition was carried out using platinum acetoacetate (Sigma Aldrich) at 350 °C under an argon atmosphere[68,69]. As-received Pt/C catalysts from Alfa Aesar (HiSpec 2000, 3000, 4000) were used as a commercial standard for electrochemical activities.

***Ex situ* catalyst physical characterization.** The prepared support material was characterized by nitrogen physisorption, analysed using BET theory (Micromeritics Instrument Corporation TriStar II), to determine the support surface area. XRD measurements were carried out on a Bruker D8 Advance diffractometer with a Cu Kα radiation source operating at 40 kV to determine the crystalline phases present in the catalyst. Rietveld refinement on the XRD patterns was completed using Bruker AXS TOPAS software, Version 4.1. TEM was performed on a Tecnai G2 electron microscope operating at 200 kV to determine particle sizes and a Tecnai F20 with a FEG operating at 200 kV to obtain images shown in Fig. 1. Elemental analyses were carried out by using a Cs probe-corrected JEOL ARM200F (cold-FEG) TEM/STEM operated at 200 kV and equipped with a Gatan GIF spectrometer. EELS data were acquired with an energy dispersion of 0.25 eV per channel. XPS was carried out by Nexus XPS service at Newcastle University using a Thermo Scientific K-Alpha instrument with an Al Kα X-ray source. The supported catalysts were characterized using TEM for Pt particle size using an average of at least 100 particle diameters sampled across three images for each catalyst; particle distances were obtained by measuring the closest neighbouring platinum particle distance for 100 particles across three TEM images for each catalyst. XRD was used to verify the Pt crystallite size, XPS to investigate the binding energies, and ICP-OES (Varian 730-ES) to confirm the Pt loading on the support. *Ex situ* advanced surface characterisations were completed using X-ray adsorption spectroscopy on the B18 line at the Diamond Light Source to investigate the morphology of Pt on the surface of the support. The Pt $L_2$ and $L_3$ edges (13,273 and 11,564 eV) were measured on beamline B18 at Diamond Light Source which operated with a ring energy 3 GeV and at a current of 300 mA. The monochromator used Si(111) crystals operating in Quick EXAFS (QEXAFS) mode. A total of three spectra were averaged for each sample. The measurements were collected using the ionization chambers in transmission mode at 298 K. Calibration of the monochromator was carried out at both edges using Pt foils. XAS was collected for the 20 wt% Pt/C and 20 eq. wt% Pt/BC catalysts only. These catalysts were prepared using boron nitride to bind pellets of the catalysts, the pellets were then placed under a $H_2$ atmosphere to reduce surface oxides. The absorption spectra were modelled with Demeter by Bruce Ravel using Ifeffit by Matt Newville[17] to solve the EXAFS equation for the first four Pt shell co-ordination numbers as described by Frenkel et al.[49] The ink formulation for the 20 eq wt% Pt/BC catalysts used for XANES consisted of 228 mg catalyst, 3.42 ml 5 wt% Nafion solution, 7.6 ml ethanol (Sigma Aldrich); a catalyst layer of 91.46 mg was coated onto a Teflon coated carbon base layer.

**Catalyst electrochemical characterization.** Catalyst inks were prepared for the electrochemical characterization techniques. The Pt/BC catalyst ink consisted of 2.5 mg of the catalyst, 0.5 ml of ethanol (Sigma Aldrich) and 25 µl of 5 wt% Nafion solution. The mixture was sonicated for 30 min and 5 µl was placed on a glassy carbon electrode. The Pt/C catalyst ink consisted of 5 mg of the catalyst, 5 ml of water (Milli-Q 18.2 mΩcm), 1.5 ml isopropanol (Kimix) and 25 µl of 5 wt% Nafion solution. The mixture was sonicated for 30 min and 10 µl was placed on a glassy carbon electrode. Prior to this, the glassy carbon electrode was polished to a mirror finish using 0.1 and 0.05 µm $Al_2O_3$ polish (Buehler). The reference electrode used was $Hg/HgSO_4$ with a platinum counter electrode. Cyclic voltammetry was conducted by cycling the potential between 0.05–1.2 V versus RHE with a scan rate of 50 mVs$^{-1}$ for 50 cycles at room temperature in an oxygen-free 0.1 M $HClO_4$ electrolyte solution to electrochemically clean the catalyst surface. The scan rate was then changed to 20 mVs$^{-1}$ for 5 cycles and the potential cycled between 0.05 and 1.2 V versus RHE at room temperature in an oxygen-free 0.1 M $HClO_4$ electrolyte solution. CO stripping was conducted by purging the electrolyte solution with CO for 20 min while holding the potential at 0.1 V versus RHE, argon was then bubbled for 20 min to rid the solution of CO, after which the potential was cycled between 0.05 and 1.2 V versus RHE. The ORR activity was measured by RDE experiments in an $O_2$ saturated 0.1 M $HClO_4$ electrolyte (70% Suprapur—Merck) using the anodic scan between 0.05 and 1.2 V versus RHE, with a scan rate of 20 mVs$^{-1}$ and rotation rate of 1,600 r.p.m. at room temperature. All scans were normalized for ECSA and corrected for ohmic drop measured by impedance spectroscopy and capacitive current in an oxygen-free electrolyte. Error margins were obtained from 2 to 4 repeats for each data point. The catalyst degradation studies were completed by cycling between 0.6 and 1.0 V versus RHE with a scan rate of 50 mVs$^{-1}$ in 0.1 M $HClO_4$ electrolyte solution for 6,000 cycles at room temperature, and periodically reverted to a cyclic voltammogram between 0.05 and 1.2 V versus RHE with a scan rate of 20 mVs$^{-1}$ to investigate the changes to the $H_{UPD}$ region. The reference electrode used was $Hg/HgSO_4$ with a gold counter electrode. The $H_{UPD}$ region was used for changes in ECSA in this instance. ORR activity and ECSA were reassessed after cycling using RDE and CO stripping following the procedures described above.

**Data availability.** All data supporting this study are openly available from the University of Southampton repository at http://dx.doi.org/10.5258/SOTON/D0067.

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

## Acknowledgements

This work was largely supported by the Royal Society in the form of a Royal Society-Newton Advanced Fellowship (P.L.; grant no. NA140367). C.J. acknowledges the University of Cape Town for financial support through a UCT PhD Mobility Grant. D.K. and G.T.S. thank the EPSRC H2FC SUPERGEN (grant no. EP/J016454/1) for financial support. G.T.S. thanks the HySA/Catalysis Programme for a postdoctoral fellowship. D.K. acknowledges support from STFC (ST/K00171X/1 and ST/N002385/1). We thank Tobias Binninger and Thomas Schmidt from Paul Scherrer Institut for work with D.K. on a theoretical framework of electronic metal-support interactions that predates and informed the experiments. Facilities use was supported by the HySA/Catalysis Centre of Competence (UCT) and the Department of Chemistry (UoS). Additionally, we thank the National EPSRC XPS Users' Service (NEXUS) at Newcastle University, Diamond Light Source, the 'South of England Analytical Electron Microscope' supported by EPSRC (EP/K040375/1), and the UCT Electron Microscope Unit for instrument access. Tarek Kollmetz assisted with powder conductivity measurements.

## Author contributions

C.J. and G.T.S. synthesized materials and performed XRD, TEM, XPS and electrochemical characterization; C.J., P.S.W., A.S.L. and D.W.I. performed XAS studies at Diamond; G.T.S. and M.C. performed EELS analyses supported by T.P.; D.K., P.L. and A.E.R. assisted in interpretation of results; C.J. and D.K. co-wrote the manuscript; all authors commented on the manuscript; D.K. conceived the experimental idea whilst being on sabbatical with P.L.'s group.

## Additional information

**Competing interests:** The authors declare no competing financial interests.

