## [Peer Review File · Nature Communications]

Reviewers' comments:

Reviewer #1 (Remarks to the Author):

Dear Sir/Madam

After careful evaluation of the manuscript, I would not recommend that this work be published in Nature Communications. Some of the reasons are summarized below:

1. The work presented is already known since 1966 by Grubb et al (Nature 1966). The enhance ORR activity of the Pt/C4B catalyst has also been reported by Haifeng Lv et al. (J. Mater. Chem., 2012,22, 9155-9160).
2. The B4C itself exhibit catalytic activity towards ORR and OER activity (Wen-Bin Luo et al, J. Mater. Chem. A, 2015,3, 18395-18399). Therefore, extensive study of the aforementioned has to be taken into consideration when evaluating the ORR activity.
3. At the XPS data, the O spectra are missing, rising questions, especially due to the presence of a clear PtO shoulder on the Pt spectra for the case of 10 eq. wt% Pt/B4C. It is widely known that the binding energy of PtO shifts to higher energies compared to metallic Pt. The fitting of the Pt spectra would have cleared this open question- if provided, as it is essential towards the claim of metal support interaction due to the observed shift in the binding energy.
4. There is a lack of information and consistency concerning their B4C support and the calculation on the equivalent Pt loading on the B4C. To be more precise, the BET area of the B4C is not reported in the manuscript. Although this is a minor issue that can easily be fixed, nevertheless by back-calculating from the Pt eq wt% on B4C and the known surface area and Pt loading of their reference catalyst the BET area of the B4C is found to be roughly 40m²/gc. But the manufacturing company gives a BET area around >75 m²/gc according to their website and product description.
5. The electrochemistry data are questionable due to the following reasons:
 - (a) If you take into consideration the error bars of the reported mass activity and specific activity values, the difference is marginal.
 - (b) The performance of the reference catalysts (Pt supported on Vulcan Carbon) that is reported in the literature [I. Takahashi et al., J. Power Sources, 195, 6312 (2010). Y. Garsany, et al., Anal. Chem., 82, 6321 (2012)]. The latter suggests poor film quality on the RDE tip [Y. Garsany, I.L. Singer, and K.E. SwiderLyons, J. Electroanal. Chem. 662 (2011) 396-406.].
 - (c) The comparison of the carbon based catalyst with literature is after extrapolating data from 60oC to 25oC from H.A Gasteiger et al. from 2005 (the data in the aforementioned paper have been characterized based on today's standards below average), while there are numerous studies since then conducted under the same conditions which could have been used as a literature benchmark.
 - (d) The Nafion is known to influence the activity of the catalyst during RDE [Kocha, S. S., et al., ECS Trans. 2012, 50.]. Since the BET surface area of B4C is not given, it is not clear by the data provided if both catalyst have the same ionomer film thickness.

Reviewer #2 (Remarks to the Author):

This paper reports on a study of Enhanced oxygen reduction activity and stability of Pt nanoparticles supported on boron carbide(BC) by purely electronic metal-support interactions. The study was motivated by the hypothesis of strong-metal-support-interaction between Pt nanoparticle and its support BC to tune the support work function. The authors used many experimental techniques to characterize the support-metal interaction. The study presented here is an example of a nicely carried out research project characterized by a close collaboration between theoretical consideration, electrochemical testing, and materials characterization. The paper is well-written. In a way, this paper breaks new ground in metal-support interaction for enhancing both catalytic activity and stability in ORR. I appreciate the intent and design of the work. However, the paper contains some major defects which need to be clarified before it can be accepted.

(1) Since the claim of the paper is that Pt supported on B4C gave enhanced catalytic ORR activity, I do not understand why they used Graphite-rich Boron Carbide composites (BC) in which B4C is only 38.7 % of the composite. Nearly pure B4C nanoparticles are commercially available.

(2) In order to calculate specific catalytic activities or ECSA, precise value of the weight percent of Pt needs to be known. I think the 10, 20, 40 eq wt % given are just nominal ones. What is the actually measured eq wt%? In paper, they said "the eq. wt % was calculated from surface areas calculated from nitrogen physisorption using a TriStar II 3020 (Micrometrics) and interpreted using Brunauer, Emmett and Teller (BET) theory." This reviewer fails to understand how can the content of Pt be determined this way.

(3) About charge transfer between metal and BC, the authors should state clearly the direction of electron transfer.

Reviewer #3 (Remarks to the Author):

The manuscript, "Enhanced oxygen reduction activity and stability of Pt nanoparticles by purely electronic metal-support interactions" by C. Jackson et al. was examined.

The manuscript presents the preparation and characterization of Pt nanoparticles on graphite-rich boron carbide. These materials were tested towards the oxygen reduction reaction in acidic media. The manuscript presents a good scientific approach with different techniques and could have noteworthy impact in the field; however, this referee found some subjects of concern regarding the experimental results – in particular, regarding the electrochemical measurements. Therefore, this version, without further clarification, is not currently suitable for publication in Nature Communication. Please note that I would certainly reconsider my recommendation after revision and discussion.

Major concerns:

1) Information regarding how the measurements of the BET were performed is required. Together with the BET curves, this information needs to be included in the SI. The authors have used these curves for the normalization of the loading which is an important part of the manuscript and which may significantly affect the results and conclusions.

2) On page 4, in the XRD section, the authors included values of composition obtained from Rietveld refinement. The authors would agree that that these values are far from being accurate, since the Rietveld refinement on this type of catalyst is not appropriate. Errors are very large (3.0 ± 1.7 wt %). Perhaps a better composition analysis could be obtained from other characterizations techniques. Also, could the authors provide the composition from the fitting of the XPS peaks from Figure 3?

3) The authors reported the values of the Pt particle size; however, I have not found any reference to the particle size of BC. This information should be included in the manuscript.

4) On page 5, the authors say that, from the Rietveld refinement, the average Pt particle size is between 3-5.4 nm. Later, in the same paragraph, the authors say that slightly larger particles are expected. However, from the TEM images and from the data in Table 1, the particle size is much smaller (below 3 nm in all the cases). The authors have not discussed these contradictory observations and results.

5) The intention of the authors (which is correct) is to rule out any other possible contribution to the catalytic activity or durability than the electronic effect. For that reason, obtaining catalysts with the same Pt particle size and inter-particle distance is very important. The scale bar of the TEM images on Figure 1A, 1B and 1C are the same, and the images indicate that the Pt nanoparticles in Figure 1A and Figure 1C are quite similar. This is reflected in the particle size distribution. However, Figure 1B shows significantly smaller nanoparticles and this is not very clear in the particle size distribution. In addition, the particle size distribution information taken from Table 1 indicates that the particle sizes from Figure 1B and 1C should be similar, but that the particle size from Figure 1A should be significantly smaller. It seems that the TEM images from Figure 1A and 1B are swapped, so the authors might want to revisit the Figures.

6) Similar to the previous comment, the authors will need to provide more information regarding the measurement of the average inter-particle distance. While commendable that the authors include error values in Table 1, the values are not very encouraging. In some cases, the average inter-particle distance has an error of almost 60 % and, in the best-case scenario, the error is 28%. This might not affect the interpretation of the electronic effect on the catalytic activity, but it certainly will affect the durability of the catalyst since it will affect the degradation mechanism. The authors will need to include a discussion on this issue.

7) The authors determine the ECSA from the oxidation of the monolayer of CO. In order to obtain the ECSA, they have used the value of 420 $\mu\text{C}/\text{cm}^2$. This is a common strategy found in the literature, but it is frequently implemented incorrectly. These values assume a compact monolayer of CO with a coverage equal to 1 with the CO sitting on a top-site on the platinum atom. However, as was extensively studied, the coverage of carbon monoxide strongly depends on the surface structure and the particle size. As was demonstrated by M. Weaver (Langmuir, 2000, 16 (2), 811–816) the coverage of CO on a perfect (111) single crystal will not be higher than 0.7. Furthermore, it has been demonstrated that the presence of defects on the surface favor the adsorption of CO on bridge-site and hollow-site; as a consequence the coverage of CO decreases. By using the value of 420 $\mu\text{C}/\text{cm}^2$, the authors could be underestimating the ECSA and therefore overestimating the surface area specific activity. This referee understands the need to use a different probe, rather than the proton adsorption, to determine the ECSA; however, the use of CO might be leading the authors to incorrect conclusions. The authors might need to determine the ECSA using other surface probes; for example, Cu UPD.

8) The authors report that the ECSA of the nanoparticles with particle size of around 2.3-2.9 nm (Table 1) is 81-69 m^2/gPt (Table 2). If we consider that the nanoparticles are spheres, nanoparticles of 2 nm should give surface areas of 139 m^2/g ; nanoparticles of 3 nm should give surface areas of 93 m^2/g (assuming the density of Pt of 21.45). I would expect a smaller value of the ECSA due to point of contact; however, the values are significantly different. This difference could be attributed to possible contamination, or to the underestimation of the ECSA described above. The authors should justify and discuss.

9) The authors have included a discussion of the durability of the catalyst based only on the ECSA (Figure 5); however, the durability issues of the PEMFC catalyst need to be addressed, not only based on the ECSA, but also on the ORR polarization curves. In order to prove the relevance of this work and its impact in the field, the authors need to provide evidence of ORR catalytic activity during the durability cycles.

10) The comment on page 18 regarding the durability of the catalyst after 30000 cycles is an unrealistic extrapolation; without proof, this comment should be removed.

Minor concern:

1.- The title is too broad and it does not reflect the content of the manuscript. Certainly, the scope of the paper is intended to attract readers to investigate the electronic metal-support interactions; however, the paper is based only on interactions between platinum and a particular carbide. The title, therefore, needs to reflect this; "Boron carbide" needs to be included in the title.

2.- In relation to my previous comment, the introduction could better reference previous studies of metal support interactions with carbide materials rather than briefly referencing the SMSI electronic effects of Pt on TiO_2 catalysts. A number of works can be found in the literature which previously reported enhancement of catalytic activity and durability towards the ORR on Pt-supported nanoparticles on carbide nanomaterials (see, for example, "Perspective Review," ACS Catal., 2013, 3 (6), 1184–1194 and cross references).

3.- The authors should include the information of the RDE. The diffusion currents reported in Figure S5 are odd; but, of course, depend on the geometrical area of the carbon disc.

4.- Also related to Figure S5, the authors should provide information about the initial potentials of the cycle and the direction of the scan. The kinetic currents taken in a positive scan with initial potential 0.1 V are significantly higher than those recorded in a negative scan with start potential 1.2 V. In the second case, the kinetic current will be strongly affected by reduction of the oxide layer formed at those high potentials.

5.- Have the ORR polarization curves been corrected by the contribution of the blank voltammetries

and the double layer capacity? Have the ORR polarization curves been corrected by the ohmic drop? This information should be included.

Rebuttal Letter

Enhanced oxygen reduction activity and stability of boron carbide supported Pt nanoparticles by electronic metal-support interactions

Colleen Jackson, Graham T. Smith, David Inwood, Andrew S. Leach, Penny S. Whalley, Mauro Callisti, Tomas Polcar, Andrea E. Russell, Pieter Levecque and Denis Kramer**

We thank all reviewers for the constructive and thorough critique of our manuscript. We give a detailed point-by-point response below. Due to the amount of information and additional experiments requested, we have adopted a tiered approach: (i) where changes/additions are most relevant for a general readership, we have adapted the manuscript, (ii) where additional information is of value to readership with specialist interests, we have provided additional supplemental material, and (iii) further detailed responses are given in this letter, which we expect to be published alongside the manuscript and supplemental information in line with journal policy. We first address points that were raised by more than one reviewer, before responding individually.

BET surface area of support and equivalent loading

R1 (4) There is a lack of information and consistency concerning their B4C support and the calculation on the equivalent Pt loading on the B4C. To be more precise, the BET area of the B4C is not reported in the manuscript. Although this is a minor issue that can easily be fixed, nevertheless by back-calculating from the Pt eq wt% on B4C and the known surface area and Pt loading of their reference catalyst the BET area of the B4C is found to be roughly 40m²/gc. But the manufacturing company gives a BET area around >75 m²/gc according to their website and product description.

R2 (2) In order to calculate specific catalytic activities or ECSA, precise value of the weight percent of Pt needs to be known. I think the 10, 20 , 40 eq wt % given are just nominal ones. What is the actually measured eq wt% ? In paper, they said “the eq. wt % was calculated from surface areas calculated from nitrogen physisorption using a TriStar II 3020 (Micrometrics) and interpreted using Brunauer, Emmett and Teller (BET) theory.” This reviewer fails to understand how can the content of Pt be determined this way.

R3 (1) Information regarding how the measurements of the BET were performed is required. Together with the BET curves, this information needs to be included in the SI. The authors have used these curves for the normalization of the loading which is an important part of the manuscript and which may significantly affect the results and conclusions.

We apologize for not explicitly reporting BET surface areas of the supports. This was an oversight. BET surface areas are now reported in the manuscript under Results and the underlying adsorption isotherms are now included in the Supplemental Information as Figure S3.

The BET surface area of Vulcan ($\sim 259 \text{ m}^2/\text{g}$) is about three times larger than the surface area of the BC support ($\sim 80 \text{ m}^2/\text{g}$), and, therefore, we have targeted an actual Pt loading per gram on the BC support that is roughly one third of the respective benchmark Vulcan support. In detail, we aim to keep the mass Platinum per surface area support constant:

$$\frac{m_{Pt}^C}{A_C} = \frac{m_{Pt}^{BC}}{A_{BC}} \quad (\text{R1})$$

The weight percent Platinum is defined as

$$x_C = \frac{m_{Pt}^C}{m_{Pt}^C + m_C} \quad \text{and} \quad x_{BC} = \frac{m_{Pt}^{BC}}{m_{Pt}^{BC} + m_{BC}} \quad (\text{R2})$$

Combining (R1) and (R2), the target weight percent Platinum on BC that yields the same mass Platinum per surface area as on C can be calculated from

$$x_{BC} = \frac{\gamma(x_C^{-1} - 1)^{-1}}{\gamma(x_C^{-1} - 1)^{-1} + 1} \quad (\text{R3})$$

with $\gamma = a_{BC}/a_C$ being the ratio of specific surface areas of both support materials. We used a value of $\gamma = 80.4/259 = 0.31$ to calculate the target weight percent on BC from the actual weight percent on C in Table S1.

We also thank Reviewer 1 for highlighting a mistake in the data contained in Table S1. The reviewer is of course right that the data originally reported in Table S1 leads to a surface area of about $40 \text{ m}^2/\text{g}$. We accidentally included data from a different support material when compiling the supplementary material. We apologize. This has, however, no further bearings on the results as all other data is based on the correct values that are now reported in Table S1.

Specific response to Reviewer 1

R1 (1) The work presented is already known since 1966 by Grubb et al (Nature 1966). The enhance ORR activity of the Pt/C4B catalyst has also been reported by Haifeng Lv et al. (J. Mater. Chem., 2012,22, 9155-9160).

The reviewer is right that early work on Pt/B₄C dates back to the 1960s and has occasionally been revisited as we have acknowledged by citing the work of Grubb et al. and Lv et al. (reference 24,25 in the original manuscript).

Our study, as presented in the original manuscript, does not focus on enhancement of ORR activity. There are many ways to achieve higher activity as the substantial body of work on Pt alloys shows. Rather, our main objective is to experimentally isolate and study electronic metal-support interactions as an alternative design approach for ORR electrocatalysts that promises higher activity without compromising durability. Our careful experimental design and the comprehensive set of characterization methods used have allowed us to achieve this objective for the first time.

It is our hope that this deeper understanding of metal-support interactions will stimulate novel rational catalyst-by-design approaches for more active and durable ORR

electrocatalysts. D-Band theory (J. Nørskov and co-workers) provided the leading design principle for advanced ORR catalysts over the last decade. Every effort was made to identify alloy partners for Pt that shift the d-band centre sufficiently to destabilise oxygen intermediates with substantial success at the lab scale [e.g., *Science* 315 (Jan 2007) 493, *Nature Chem.* (2009) 552, *Nature Mater.* 6 (2007) 241, *Science* 352 (Apr 2016) 73]. However, it becomes increasingly clear that stability of the alloys could prove insufficient to transition those approaches into applications. We show that purely electronic effects across the metal-support interface can positively affect activity and durability without changing the catalyst particles themselves. This is important because it runs contrary to the widely-held belief that electronic effects are irrelevant beyond one or two atomic layers away from the interface due to the effective charge screening of metals. These conclusions have been reached from Density-Functional-Theory calculations, which predict full screening of electronic effects for metallic adlayers within a few Angstroms away from the catalyst-support interface with negligible impact for catalytic properties at the relevant nanometre scale [*ChemCatChem* 4 (2012) 228, *Surf. Sci.* 138 (1984) 84]. Our work provides experimental proof that electronic support effects, which were previously believed to be relevant only for sub-nanometre clusters [e.g., *Angew. Chem.* 42 (2003) 1297, *Appl. Surf. Sci.* 257 (2011) 6607, *JACS* 134 (2012) 8968], expand to technologically important nanoparticles.

R1 (2) The B4C itself exhibit catalytic activity towards ORR and OER activity (Wen-Bin Luo et al, *J. Mater. Chem. A*, 2015,3, 18395-18399). Therefore, extensive study of the aforementioned has to be taken into consideration when evaluating the ORR activity.

The work of Luo et al. cited by the referee concerns the oxygen electrochemistry in alkaline and non-aqueous electrolytes, and those results cannot directly be translated to the acidic system studied here. This distinction, however, is important, and we have added “in acidic media” to the abstract for clarity. In acidic electrolyte, the uncatalysed BC support shows no activity towards ORR as can be seen from the RDE measurements shown in Figure R1.

Figure R1: ORR activity of uncatalysed BC compared to 40 eq. wt% Pt/BC; the anodic scan was recorded at 1600 rpm between 0.05 V and 1.2 V vs RHE, all experiments performed in 0.1 M HClO₄ at room temperature recorded at 20 mV/s.

R1 (3) At the XPS data, the O spectra are missing, rising questions, especially due to the presence of a clear PtO shoulder on the Pt spectra for the case of 10 eq. wt% Pt/B4C. It is widely known that the binding energy of PtO shifts to higher energies compared to metallic Pt. The fitting of the Pt spectra would have cleared this open question- if provided, as it is essential towards the claim of metal support interaction due to the observed shift in the binding energy.

We now report the O1s spectra in the supplemental material. The B₂O impurity in the BC support slightly complicates the interpretation of the O1s spectra for the Pt/BC catalysts, because the O1s spectra are a linear combination of contributions from the BC support and from oxygen species at the Pt surface. As far as fitting of the strongly overlapping features allows to extract quantitative information, the contribution to the overall O 1s spectrum of Pt-O species increases linearly with Pt loading as expected.

Figure R2: Fitting of the Pt 4f XPS spectra and comparison of the Pt₀ peak positions showing the same shift by 0.6 eV to higher binding energies as obtained from the overall spectrum.

It is certainly true that the Pt 4f spectrum contains contributions from oxidized Pt species. But the Pt 4f signal shifts in its entirety due to the stronger dipole field at the metal-support interface. Further, the peak positions that we used as part of our main argument are dominated by the Pt₀ contributions, as evidenced by the sharpness of these features compared to contributions from oxidized Pt species. We have fitted the Pt 4f spectra in Figure R2 to show the very good agreement between the peak positions of the fitted Pt₀ contributions (blue) and the overall peak positions. The shift of 0.6 eV between Pt/BC and Pt/C that results from comparing the Pt₀ peaks is the same that was inferred in the manuscript from comparing the overall spectra.

R1 (4) Answered above

R1 (5a) If you take into consideration the error bars of the reported mass activity and specific activity values, the difference is marginal.

We used gray shades in Figure 6 in the manuscript to indicate the range of expected SA values for both catalysts as a function of particle size. The average SA values for the Pt/BC catalysts are almost 100 $\mu\text{A}/\text{cm}_{\text{Pt}}^2$ larger than what is obtained for the same Pt particle morphology and density on Vulcan XC-72R. Given that even the most active benchmark Pt/C catalyst

shows an average SA of just above $200 \mu\text{A}/\text{cm}_{\text{Pt}}^2$, the improvement is not marginal. Please also note that we have opted to base the discussion mostly on an activity definition that does not correct for O_2 mass transport for reasons described in response to R1(5d). The O_2 corrected kinetic currents (which are also reported in Table 3) increase more than two-fold from $3.3 \text{ A}/\text{m}_{\text{Pt}}^2$ to $7.1 \text{ A}/\text{m}_{\text{Pt}}^2$ between the 40 wt% Pt/C and 40 eq. wt% Pt/BC, respectively.

R1 (5b) The performance of the reference catalysts (Pt supported on Vulcan Carbon) that is reported in the literature [I. Takahashi et al., *J. Power Sources*, 195, 6312 (2010). Y. Garsany, et al., *Anal. Chem.*, 82, 6321 (2012)]. The latter suggests poor film quality on the RDE tip [Y. Garsany, I.L. Singer, and K.E. SwiderLyons, *J. Electroanal. Chem.* 662 (2011) 396-406.].

We disagree with the reviewer on this point. All measurements are taken with multiple repeats using accepted standard methods in the field, and the results obtained fall within accepted ranges. Having applied the same methodology to both the Pt/C and Pt/BC does not change the relative improvement in performance of the Pt/BC against the Pt/C.

When comparing with the references suggested by the reviewer, it is important to note that most of the catalysts reported in those papers are manufactured by TKK. It is well known that these catalysts perform differently than the Vulcan-based HiSPEC™ series that Johnson-Matthey offers through Alfa-Aesar under RDE conditions. We have accounted for this by replacing “state-of-the-art” with “commercial” in the abstract when referring to the Pt/C benchmark catalyst.

Figure R3: Comparison of kinetic currents reported for Pt/C(Vulcan) ORR catalysts using stationary drying (Refs. [66,67] in the manuscript) with our stationary dried commercial Pt/C benchmarks (yellow).

Stationary drying is to date still the standard way of film drying for RDE in many laboratories around the world and although it might not yield the highest absolute activity, it gives consistent and reliable results across a range of catalysts. The results that we can reliably compare with from the reviewers suggestions are the 3rd and 4th entry in Table 1 in *J.*

Electroanal. Chem. 662 (2011) 396-406 and the 1st-3rd and 5th entry in Table 1 in *Anal. Chem.*, 84, 6321 (2012). If one considers that we tested at a slightly lower temperature (25 °C as opposed to 30 °C) our results correspond with the results those authors obtained for what they deemed high quality stationary dried films. This becomes apparent from Figure R3, where we compare the surface specific kinetic currents from those literature sources (adjusted for temperature according to Ref. [15]) with our stationary dried Pt/C films. We feel this is a clear indication that the methods we use reliably produce high quality films in line with literature.

R1 (5c) The comparison of the carbon-based catalyst with literature is after extrapolating data from 60°C to 25°C from H.A Gasteiger et al. from 2005 (the data in the aforementioned paper have been characterized based on today's standards below average), while there are numerous studies since then conducted under the same conditions which could have been used as a literature benchmark.

Following accepted good practice, we have opted to use a well-established commercial catalyst as internal baseline. Nonetheless, it is important to show that the used internal standard agrees with literature data within reason to show that lab procedures are adequate. We originally opted to compare against the Gasteiger data, because it is with over 3000 citations to date the most widely accepted benchmark for carbon-supported Pt ORR catalysts. We, however, accept that advances in ink and film preparation should be taken into account. We, therefore, have followed the reviewers suggestion and have used appropriate references suggested under R1 (5b) instead in the revised Figure 6 in the manuscript (former Figure 7). This also reduced the relative weight of the temperature correction as the suggested data was obtained at 30 °C rather than 60 °C. Some additional data treatment, however, was needed due to varying definitions of corrections to the specific activity. Corresponding detail has been added to the Supplementary Material as Table S3. Please also note that the applied temperature correction originated from an independent source [*J. Electrochem. Soc.*, 2006 153(10):A1955].

R1 (5d) The Nafion is known to influence the activity of the catalyst during RDE [Kocha, S. S., et al., *ECS Trans.* 2012, 50.]. Since the BET surface area of B4C is not given, it is not clear by the data provided if both catalyst have the same ionomer film thickness.

It is correct that thick electrodes and thick layers of Nafion ionomer can lead to additional losses due to mass transport limitations, protonic resistances as well as blocking/adsorption by species (e.g., by specific adsorption of sulphuric acid groups within the ionomer) and, therefore, reduced apparent surface specific ORR activity (SA) and mass specific ORR activity (MA). Our experimental design considerations were dominated by an attempt to make morphologically as similar electrodes as possible. Some compromises, however, were unavoidable due to the inherent characteristics of the different support materials and the need to make good quality inks. As we have now clarified, the BET surface area of the BC support is roughly 1/3 of the C supports. This led to thicker Pt/BC electrodes as we prioritized similar Pt particle density (and size) on the support surface and overall Pt loading over electrode thickness. It is important to note that even if the assumption of negligible electrode thickness that is used in the interpretation of RDE measurements is violated this way, one

would expect worse and not better apparent SA and MA for the Pt/BC catalysts due to increased transport resistance through the thicker electrodes.

As described in the methods section, we have prepared inks by adding 25 μ l 5wt% Nafion solution to all inks. We, therefore, held the total Nafion content in the inks constant across catalysts. This translates into higher ionomer content per support mass and surface area in the Pt/BC catalysts than for the Pt/C catalysts (ionomer/carbon ratio of about 0.2 for the Pt/C and 0.4 for the Pt/BC catalysts; \sim 6x higher Nafion weight per support surface area for Pt/BC). Following the logic of Kocha *et al.*, one would, therefore, expect a detrimental impact on SA and MA for the Pt/BC catalysts if the different Nafion contents result in appreciably different transport losses and/or blocking/adsorption phenomena. Differences in Nafion content can, therefore, not explain the higher activity seen for the Pt/BC catalysts.

Please note that mass transport limitations are to some extent accounted for by calculating kinetic currents i_k in Tables 3 and 4, following standard practice. Comparison of specific activities (without O₂ mass transport correction) and kinetic currents in Table 3, shows that the Pt/BC catalysts compare even more favorable if kinetic current is used as the metric, in line with our arguments above. We, however, consider specific activities without O₂ mass transport correction a more direct, entirely experimentally defined, metric (i.e., less assumptions are needed). For instance, amongst other things [Nano Research 7(1) (2014) 71-78], the standard approach to compensate for O₂ mass transport limitations using $1/j = 1/j_k + 1/j_{lim}$ implicitly assumes a reaction order of one for the ORR, while experimental evidence suggests values could be closer to 0.75 [J. Power Sources 195 (2010) 6312], especially for ionomer covered Pt surfaces [J. Electrochem. Soc. 144 (1997) 2973]. This leads to significant correction uncertainty and possibly inflated kinetic currents for highly active catalysts where the geometric current density approaches or indeed exceeds 1/2 of the limiting current density at 0.9 V. We have, therefore, chosen to base the discussion mostly on specific activities without O₂ mass transport correction as a more conservative representation of the ORR activity improvement of the Pt/BC catalysts over the Pt/C catalysts.

Specific response to Reviewer 2

R2 (1) Since the claim of the paper is that Pt supported on B₄C gave enhanced catalytic ORR activity, I do not understand why they used Graphite-rich Boron Carbide composites (BC) in which B₄C is only 38.7 % of the composite. Nearly pure B₄C nanoparticles are commercially available.

Pure B₄C is a semi-conductor with a band gap of about 2.1eV. Stoichiometric B₄C, therefore, shows insufficient electrical conductivity as an electrocatalyst support. Figure R4 shows CVs of high purity B₄C (obtained commercially from Superior Graphite) and the same B₄C catalyzed with different amounts of Pt nanoparticles, along with voltammetry from the electrode disk without any catalyst deposited. None of the materials show any significant double layer capacitance or any characteristics of Pt. This strongly indicates that Pt supported on pure B₄C is electrochemically inactive due to insufficient electrical conductivity of the support.

Figure R4: Cyclic voltammetry of pure B_4C and pure B_4C catalyzed with Pt; characteristic Pt features are missing indicating that Pt is not electrochemically active on pure B_4C .

Figure R5: Electronic structure and thermochemistry of B-C site substitutions in B_4C as calculated using Density Functional Theory (DFT).

Intrinsic doping is attractive to introduce charge carriers for increased conductivity. We, therefore, initially studied the energetics of B-C site substitutions close to stoichiometric B_4C using Density-Functional-Theory (DFT). Figure R5 condenses the results. The calculated band structure of stoichiometric B_4C is shown to the left, reproducing the semi-conducting characteristics of B_4C well. The convex hull plot to the right in Figure R5 shows that boron-rich B_4C is thermodynamically feasible (i.e., B_{12} -CBC is located on the convex hull of thermodynamic ground-states and features in the B-C phase diagram), but we were unable to identify a site substitution that leads to a thermodynamically competitive carbon-rich configuration. We concluded that it should be possible to increase electric conductivity by intrinsic p-type doping (i.e., B-rich B_4C), but very limited via intrinsic n-type doping (i.e., C-rich B_4C). However, p-type doping would have introduced holes in the valence band and lowered the Fermi level (and therefore likely increased the work function) of B_4C , a situation that we tried to avoid as we were targeting a small support work function for pronounced electronic effects. We, therefore, settled for a carbon-rich composite approach where we aimed for some limited n-type doping within the B_4C phase to achieve a high Fermi level and sufficient

electrical conductivity at the nanoscale, supported by a substantial phase fraction of highly conductive carbon as conductivity additive.

R2 (2) Answered above.

R2 (3) About charge transfer between metal and BC, the authors should state clearly the direction of electron transfer.

This is a very interesting question and more complicated to answer than one might think at first thought. The XPS results shown in Figure 3 taken by themselves indicate a stronger electron transfer from support to catalyst as illustrated in Figure 3B. The support has a smaller work function (literature suggests values of the order of 4-4.5eV) than the Pt particles (~5.5eV). Hence, electron transfer occurs from the support to the particle to equilibrate the respective Fermi levels.

One must, however, acknowledge that the interface in general is charged under electrochemical conditions, while XPS probes the catalysts under UHV conditions and overall neutrality. This raises the question how the electronic effects seen by XPS under UHV conditions transfer to electrochemical environments. We have been able to collect *in-situ* XANES data while the manuscript was under review that (i) provide direct evidence that electronic effects prevail under electrochemical conditions at relevant potentials and (ii) show that overall Pt particles are more positively charged if supported on BC at ORR relevant potentials. Following Bockris *et al.* [J. Chem. Phys. 49 (1968) 5133], this can be rationalized by realizing that work function and potential of zero charge (pzc) are proportional to each other according to

$$E_{q=0} = \Phi - \chi^s + \delta\chi - const. \quad (R4)$$

up to a constant defining the potential scale. Where, Φ is the work function and χ^s and $\delta\chi$ are the surface potential and its sensitivity towards the electrolyte environment, respectively. According to equation R4, the smaller work function of BC-based electrodes causes a negative shift of pzc and, hence, more positively charged electrodes at the same potential.

We have added these insights to the manuscript. Figure 3 now also shows *in-situ* XANES data, and we added one paragraph to the results section describing the XANES results and one paragraph to the discussion explaining the relation between XPS, XANES, work function and pzc. The added paragraphs read:

In-Situ X-Ray Absorption Near Edge Structure (XANES). Figure 2C shows the XANES Pt L_2 and L_3 edges for the 20 wt% Pt/C and 20 eq. wt% Pt/BC catalysts measured under potentiostatic control in electrochemical environment. The increase in the L_2 white line observed for the Pt/BC catalyst indicates an increase in unoccupied d-states above the Fermi level for Pt/BC^[18] in electrochemical environments. Using standard post-processing^[13], d-band occupancy can be estimated to be 63% for Pt/C and 58% for Pt/BC at 0.744 V vs. NHE.

Charge transfer and d-band occupancy. We have argued above that the lower work function of B_4C leads to stronger charge transfer across the interface. Because the work function of B_4C (~4-4.5 eV) is smaller than the work function of C (~5 eV) and Pt (~5.5 eV), stronger electron transfer must occur from the support to the Pt particle to equilibrate the respective Fermi levels as illustrated in Figure 2B. Hence, the stronger polarisation of the metal-support interface measured by XPS corresponds to more negatively charged Pt particles supported on BC under UHV conditions.

The situation in electrochemical environment, however, is more complex. The in-situ XANES spectra shown in Figure 2C clearly shows an increased hole density in the Pt d-band manifold when particles are supported on BC (i.e., more positively charged particles). This apparent contradiction can be resolved by realising that in-situ XANES probes a charged interface under potentiostatic control, while ex-situ XPS necessarily probes the catalyst under overall neutrality. Because work function and potential of zero charge (pzc) are proportional to each other^[10], the smaller support work function of the BC-based catalysts will cause a negative shift of the pzc and, hence, a more positively charged Pt/BC catalyst if held at the same potential as the Pt/C catalyst in the same aqueous electrolyte. Finally, it is noteworthy that the support-induced decrease in d-band filling seen for Pt/BC is reminiscent of electronic effects seen for highly active Pt-alloys^[14].

Specific response to Reviewer 3

R3 (1) Answered above

R3 (2) On page 4, in the XRD section, the authors included values of composition obtained from Rietveld refinement. The authors would agree that that these values are far from being accurate, since the Rietveld refinement on this type of catalyst is not appropriate. Errors are very large (3.0 ± 1.7 wt %). Perhaps a better composition analysis could be obtained from other characterizations techniques. Also, could the authors provide the composition from the fitting of the XPS peaks from Figure 3?

It is generally difficult to accurately characterize the stoichiometry of inorganic compounds containing boron and carbon as their low atomic numbers lead to reduced sensitivity in many physical techniques and their chemical inertness makes chemical approaches difficult. We have obtained independent confirmation of the elemental composition of the BC support using EELS, obtained at five data points at each of five different locations. EELS analysis gave a mean B:C ratio of 39:61, which translates into a phase fraction of B_4C of about 48% if the minority impurity B_2O is disregarded. The maximum (77:23) and minimum (19:81) observed spot B:C ratios further provided secondary evidence for a two-phase composite material composed of a carbon-rich (i.e., graphitic carbon) and a boron-rich phase (i.e., B_4C). The surface sensitivity of XPS makes it less suitable as a probe for bulk composition. However, the peaks attributed to B_4C in Figure 3 account for 45% of the adsorption in the C1s spectrum.

EELS and XPS analysis are in reasonable agreement with the composition estimate from Rietveld refinement, which gave a B_4C phase fraction of 42% (i.e., 38.7 wt% B_4C). Taken together, there is sufficient evidence for the two-phase composite nature of the BC support, which comprises about 45% B_4C .

We have adapted the manuscript and included the EELS result. The relevant section now reads:

Using Rietveld refinement, the spectrum indicates 58.0 ± 7.7 wt% graphite, 38.7 ± 7.2 wt% B_4C and 3.0 ± 1.7 wt% B_2O in the sample. This is in reasonable agreement with the average B:C ratio of 39:61 as obtained from EELS.

R3 (3) The authors reported the values of the Pt particle size; however, I have not found any reference to the particle size of BC. This information should be included in the manuscript.

The BC particle size was mentioned when presenting the TEM results:

The support particles are homogenous with an average particle size of approximately 40 nm, and Pt nanoparticles are well dispersed on the BC support with average particle sizes ranging from 2.3 – 2.9 nm.

As we have updated TEM images in response to R3(5) below, we took care to obtain images that also provide a visual impression of typical support particles size.

R3 (4) On page 5, the authors say that, from the Rietveld refinement, the average Pt particle size is between 3-5.4 nm. Later, in the same paragraph, the authors say that slightly larger particles are expected. However, from the TEM images and from the data in Table 1, the particle size is much smaller (below 3 nm in all the cases). The authors have not discussed these contradictory observations and results.

We have refined our methodology to obtain Pt crystallite sizes from XRD. While we previously used values from Rietveld refinement of the entire spectrum, we now use a fit of the Scherrer equation to specific Pt peaks. The relevant section in the manuscript now reads:

The average Pt crystallite size was confirmed by a fit of the Scherrer equation to the Pt(200), Pt(220) and Pt(311) peaks at 46° , 67° and 81° in the XRD spectra, respectively. This analysis yielded a volume-weighted average particle size of approximately 3.9 nm for the various loadings, in good agreement with the size measured via TEM (Table 1).

It is not uncommon—especially in the catalysis literature where particle sizes close to the detection limit of about 2 nm with substantial spread are common—for average particle sizes from XRD to be larger than from TEM (e.g., *Appl. Catalysis A* 2008, 2:187-195). A fit of the Scherrer equation to XRD spectra yields a volume-weighted average, because larger particles are stronger individual scatterer. Hence, larger particles are stronger weighted than smaller particles in the XRD spectrum, which leads to a larger average than the geometric average obtainable from TEM [*J. Appl. Crystallogr.*, 33(6), 1335–1341 (2000)].

R3 (5) The intention of the authors (which is correct) is to rule out any other possible contribution to the catalytic activity or durability than the electronic effect. For that reason, obtaining catalysts with the same Pt particle size and inter-particle distance is very important. The scale bar of the TEM images on Figure 1A, 1B and 1C are the same, and the images indicate that the Pt nanoparticles in Figure 1A and Figure 1C are quite similar. This is reflected in the particle size distribution. However, Figure 1B shows significantly smaller nanoparticles

and this is not very clear in the particle size distribution. In addition, the particle size distribution information taken from Table 1 indicates that the particle sizes from Figure 1B and 1C should be similar, but that the particle size from Figure 1A should be significantly smaller. It seems that the TEM images from Figure 1A and 1B are swapped, so the authors might want to revisit the Figures.

Although all images show a scale bar representing 20nm, images were taken with slightly different magnifications. A larger magnification was used for Figure 1C than for Figures 1A and 1B. We have tried to highlight this visually in Figure R6, where we have magnified the scale bars of the original Figure 1B and 1C three-fold and indicated the different lengths by red lines. The original Figure 1C, therefore, gives the visual impression of slightly bigger particles due to a larger magnification.

Figure R6: Comparison of magnifications used in the TEM images in Figure 1.

We have retaken TEM images of all catalysts at a consistent magnification of 270k and updated Figure 1 in the manuscript accordingly. The displayed histograms are still based on the larger original set of images (three images for each catalyst).

R3 (6) Similar to the previous comment, the authors will need to provide more information regarding the measurement of the average inter-particle distance. While commendable that the authors include error values in Table 1, the values are not very encouraging. In some cases, the average inter-particle distance has an error of almost 60 % and, in the best-case scenario, the error is 28%. This might not affect the interpretation of the electronic effect on the catalytic activity, but it certainly will affect the durability of the catalyst since it will affect the degradation mechanism. The authors will need to include a discussion on this issue.

The platinum inter-particle distance distribution was obtained by measuring the closest neighbouring platinum particle distance for 100 particles across three TEM images for each catalyst.

We have added the relevant information to the Methods section. The relevant part now reads:

The supported catalysts were characterised using TEM for Pt particle size using an average of at least 100 particle diameters sampled across three images for each catalyst; particle distances were obtained by measuring the closest neighbouring platinum particle distance for 100 particles across three TEM images for each catalyst.

Please note that the values given in Table 1 are not measurement errors. They are standard deviations characteristic of the inherent distribution of particle sizes and inter-particle distances as far as this can be gauged from TEM. We agree that the size distribution and particle dispersion of Pt on the supports should be as similar as possible for comparability. For the 40 wt% Pt/C and 40 eq. wt% Pt/BC catalysts that were used for the durability studies this is clearly the case with particle sizes of $2.7 \pm 0.7\text{nm}$ ($2.6 \pm 0.6\text{nm}$) and inter-particle distances of $5.35 \pm 1.6\text{nm}$ ($5.09 \pm 1.3\text{nm}$) for Pt/C (Pt/BC), respectively.

We have clarified this by expanding the TEM results section. The relevant sentence now reads:

Pt particle size distribution and dispersion for each catalyst are quantified and contrasted in Table 1, which shows average particle size with standard deviation as well as average inter-particle distances with standard deviation.

Please note that although the initial morphologies are very similar, there are nonetheless differences in the degradation modes on both supports stipulated by the different metal-support interactions as we explain in detail when responding to your comment (9) below.

R3 (7) The authors determine the ECSA from the oxidation of the monolayer of CO. In order to obtain the ECSA, they have used the value of 420 $\mu\text{C}/\text{cm}^2$. This is a common strategy found the literature, but it is frequently implemented incorrectly. These values assume a compact monolayer of CO with a coverage equal to 1 with the CO sitting on a top-site on the platinum atom. However, as was extensively studied, the coverage of carbon monoxide strongly depends on the surface structure and the particle size. As was demonstrated by M. Weaver (Langmuir, 2000, 16 (2), 811–816) the coverage of CO on a perfect (111) single crystal will not be higher than 0.7. Furthermore, it has been demonstrated that the presence of defects on the surface favor the adsorption of CO on bridge-site and hollow-site; as a consequence the coverage of CO decreases. By using the value of 420 $\mu\text{C}/\text{cm}^2$, the authors could be underestimating the ECSA and therefore overestimating the surface area specific activity.

This referee understands the need to use a different probe, rather than the proton adsorption, to determine the ECSA; however, the use of CO might be leading the authors to incorrect conclusions. The authors might need to determine the ECSA using other surface probes; for example, Cu UPD.

This is a very valid concern. While we are less concerned about the absolute ECSAs for this comparative study, it is in principle possible that the different charge state of the Pt particles on the compared supports stabilizes a different adsorption mode of CO and/or leads to

different covalency of the CO-Pt bonds, which could lead to a systematic underestimation of the ECSA of Pt/BC relative to Pt/C from CO stripping and, in consequence, to an overestimation of relative surface specific activity.

Figure R7: Comparison of ECSAs as determined by CO stripping and Cu-UPD; Cu-UPD tends to yield lower ECSA values across both types of supports.

We followed the suggestion of using Cu-UPD as an independent method to test for systematic errors in determining ECSA via CO stripping. Figure R7 compares ECSAs as obtained from CO stripping and Cu-UPD, following the method suggested by Green and Kucernak [*J. Phys. Chem. B*, 2002 106:1036-1047]. The Cu-UPD charge was determined by subtracting background CVs obtained in the supporting electrolyte from the Cu-UPD CVs and integration of the positive current peaks over the relevant voltage range. The Cu-UPD CV of the 40 wt% Pt/C and the 40 eq. wt% Pt/BC catalysts are shown to the right as examples, respectively. The respective CO stripping voltammetry can be found in the supplemental information.

As Figure R7 shows, Cu-UPD confirms the ECSAs obtained for Pt/C using CO stripping for the relevant particle sizes (2-3 nm range). Unfortunately, we were unable to extract an ECSA from Cu-UPD for the 10 eq wt% Pt/BC (i.e., only 2.7 wt% Pt actual) with precision because of the relative size of the faradaic signal to the capacitive background. The data for the 20 and 40 eq. wt% Pt/BC, however, seems to suggest that Cu-UPD yields slightly *smaller* ECSAs for the Pt/BC catalysts compared to CO stripping although results are within the measurement error. Cu-UPD, therefore, gives no indication for a systematic underestimation of the Pt/BC ECSAs compared to Pt/C from CO stripping.

We, however, find Cu-UPD to be slightly less reproducible and more sensitive to particle size as the significant deviation for the 10 wt% Pt/C catalyst shows. We have, therefore, opted to retain the ECSAs as determined by CO stripping within the manuscript, in line with standard practice in the field.

R3 (8) The authors report that the ECSA of the nanoparticles with particle size of around 2.3-2.9 nm (Table 1) is 81-69 m²/gPt (Table 2). If we consider that the nanoparticles are spheres, nanoparticles of 2 nm should give surface areas of 139 m²/g; nanoparticles of 3 nm should give surface areas of 93 m²/g (assuming the density of Pt of 21.45). I would expect a smaller

value of the ECSA due to point of contact; however, the values are significantly different. This difference could be attributed to possible contamination, or to the underestimation of the ECSA described above. The authors should justify and discuss.

Our measured ECSAs agree well with manufacturer specifications from Alfa Aesar and previous literature [Garsany et al., J. Electroanal. Chem. 662 (2011) 396-406] for the commercial Pt/C catalysts. Table R1 compares our ECSAs with these independent measurements.

Table R1: Electrochemical Surface Areas (ECSAs) for the commercial benchmark catalysts as reported here and in the literature.

Catalyst	Our Data	Alfa Aesar	Garsany et al.
10% Pt/C	115 ± 15	105	-
20% Pt/C	82.8 ± 2.5	90	-
40% Pt/C	56.0 ± 3.6	60	49

As our results for ECSAs of the commercial benchmark catalysts agree with independent values from Alfa Aesar and literature, we see no indication of unusual contamination diminishing the measured surface areas. Given that Cu-UPD confirmed the electrochemically active surface areas for Pt/C and yielded slightly lower ECSAs for Pt/BC, we also see no indication for a systematic underestimation of the ECSAs from CO stripping. However, it is of course possible that not the entire Pt surface area is electrochemically active. For instance, a fraction of Pt particles could be electrochemically inactive due to the complex, porous morphology of the electrodes.

R3 (9) The authors have included a discussion of the durability of the catalyst based only on the ECSA (Figure 5); however, the durability issues of the PEMFC catalyst need to be addressed, not only based on the ECSA, but also on the ORR polarization curves. In order to prove the relevance of this work and its impact in the field, the authors need to provide evidence of ORR catalytic activity during the durability cycles.

This is of course a very valid point. We have now included evidence for ORR activity before and after cycling. We have abstained from periodically obtaining ORR data during the cycling because this would significantly change the aging protocol and, therefore, complicate interpretation.

Table 4 in the revised manuscript compares kinetic data of the pristine and cycled catalysts, the underlying RDE data is given in Figure S10 in the Supplemental, and we have added two paragraphs to the manuscript, one in the results section explaining the data shown in the new Table 4 and one in the discussion section that explains the different trends in ORR activity seen for the Pt/BC and Pt/C catalysts. We also measured ECSA via CO stripping before and after cycling and report the corresponding stripping CVs in the supplemental material (Figure S9).

The altered/added paragraphs read:

Results: ORR activity of cycled catalysts as obtained from RDE measurements (Figure S10) is compared with ORR activity of the pristine catalysts in Table 4 for one set of catalysts. For consistency, specific and kinetic activities are based on ECSAs measured again after 6000 cycles by CO stripping (cf. Figure S9). The ECSAs reported in Table 4 agree well with the cyclic H-UPD measurements (cf. Figure 4), showing a significantly larger loss in surface area for the Pt/C catalyst (23% loss after 6000 cycles) than for the Pt/BC catalyst (12% loss after 6000 cycles). Accordingly, mass activity of the cycled Pt/BC catalyst remains with $168 \text{ A/g}_{\text{Pt}}$ about 30% higher than seen for the cycled Pt/C catalyst. Relative loss of mass, specific and kinetic activity of the Pt/BC catalyst are in reasonable agreement with the loss in ECSA for the Pt/BC catalyst. The Pt/C catalyst, however, shows a markedly different behaviour. While mass activity also degrades, the relative loss in mass activity (9% loss) is smaller than the reduction in ECSA (23% loss). Correspondingly, surface specific and kinetic activities increase for reasons discussed later. Nonetheless, kinetic activity of the cycled Pt/C catalyst remains significantly lower than seen for the Pt/BC catalyst after 6000 cycles.

Discussion: Since particle sizes and shapes of both catalysts are initially similar, this cannot be due to a more stable particle size as starting point. Hence, the results suggest that the stronger electronic metal-support interaction between Pt and the BC support leads to a reduction in platinum dissolution and/or agglomeration. The ORR activities of the cycled catalysts reported in Table 4 suggest that mostly Pt agglomeration is inhibited for Pt/BC relative to the Pt/C catalyst. While the relative loss of mass specific ORR activity after 6000 cycles is in line with the loss of ECSA for Pt/BC, Pt/C shows an increase in surface specific and kinetic activity after cycling. This increase in surface specific activity and simultaneous loss in mass activity is a clear indication of increasing Pt particle sizes on the Pt/C support. CO stripping voltammetry (cf. Figure S9) provides further evidence for increased Pt agglomeration on the cycled Pt/C. The pre-peak that is observed for both pristine catalysts has been attributed previously to Pt agglomerates^[50]. This pre-peak feature increases for the cycled Pt/C catalyst while it disappears for the cycled Pt/BC catalyst, indicating increased particle agglomeration for cycled Pt/C.

R3 (10) The comment on page 18 regarding the durability of the catalyst after 30000 cycles is an unrealistic extrapolation; without proof, this comment should be removed.

The extrapolation has been removed from the statement. It now reads:

Due to the 1/3 reduction in loss rate per decade, the Pt/BC catalyst retains more than 90% ECSA for about 2000 cycles, while the Pt/C catalyst shows a 10% loss in surface area already after about 200 cycles.

Minor concern

R3 (M1) The title is too broad and it does not reflect the content of the manuscript. Certainly, the scope of the paper is intended to attract readers to investigate the electronic metal-support interactions; however, the paper is based only on interactions between platinum and a particular carbide. The title, therefore, needs to reflect this; "Boron carbide" needs to be included in the title.

We have adapted the title to better reflect the breadth of experimental evidence. It now specifies “boron carbide supported Pt nanoparticles”.

R3 (M2) In relation to my previous comment, the introduction could better reference previous studies of metal support interactions with carbide materials rather than briefly referencing the SMSI electronic effects of Pt on TiO₂ catalysts. A number of works can be found in the literature which previously reported enhancement of catalytic activity and durability towards the ORR on Pt-supported nanoparticles on carbide nanomaterials (see, for example, “Perspective Review,” ACS Catal., 2013, 3 (6), 1184–1194 and cross references).

We thank the reviewer for highlighting the perspective in ACS Catalysis, which is a valuable entry point into the literature surrounding carbides as support materials. We agree that an exhaustive review of the literature around carbides would add value to the manuscript. Space in the manuscript, however, is severely constrained by Journal guidelines. We have, therefore, opted to shorten the section on oxides and use the freed space to point to the perspective on carbides in ACS Catalysis as well as another review article discussing carbide supports in a wider context. We have also taken the opportunity to briefly mention why we have not considered the widely investigated TMCs such as Tungsten Carbide (i.e., insufficient stability against oxidation and small surface areas).

Finally, we would like to point out that the suggested ACS Catalysis review—amongst other things—reiterates DFT results that seem to support the widely held belief that electronic effects are screened within the first monolayer (see discussion of Pt ML on WC in the review). These conclusions were drawn from theoretically studying metallic adlayers, and our results are a clear indication that they do not hold for supported nanoparticles opening up entirely new rational design approaches as we have described in response to R1(1).

R3 (M3) The authors should include the information of the RDE. The diffusion currents reported in Figure S5 are odd; but, of course, depend on the geometrical area of the carbon disc.

We thank the reviewer for highlighting this. The presentation of the RDE data in Figure S8 (former Figure S5) has been revised and now shows currents normalized to the actual geometric surface area to show that the limiting currents of 6 mA/cm² agree with expectations for fully O₂ saturated 0.1 M HClO₄ electrolyte solutions [Electrochim. Acta, 53(7) 3181-3188 (2008)]. Please note that we commented on the abnormal limiting current of the 10 wt% Pt/C benchmark in the manuscript:

An exception is seen for the 10 wt% Pt/C catalyst, which exhibited large experimental scatter and anomalously small limiting currents during RDE measurements (Figure S8A) indicative of various detrimental effects.^[39] The 10% Pt/C catalyst is, therefore, not considered a suitable benchmark for our purposes.

Please see below comment in response to R3 (M5) regarding experimental detail for the RDE measurements.

R3 (M4) Also related to Figure S5, the authors should provide information about the initial potentials of the cycle and the direction of the scan. The kinetic currents taken in a positive scan with initial potential 0.1 V are significantly higher than those recorded in a negative scan with start potential 1.2 V. In the second case, the kinetic current will be strongly affected by reduction of the oxide layer formed at those high potentials.

The reviewer is correct that activity measurements are sensitive to scan direction due to transient surface oxides. We have consistently opted to compare positive going scans in line with common practice in the literature. Please also see next comment in response to R3 (M5).

R3 (M5) Have the ORR polarization curves been corrected by the contribution of the blank voltametries and the double layer capacity? Have the ORR polarization curves been corrected by the ohmic drop? This information should be included.

Experimental conditions for the RDE measurements are provided in the Methods section at the end of the manuscript under "Catalyst Electrochemical Characterisation". The relevant part reads:

The ORR activity was measured by rotating disc electrode experiments in an O₂ saturated 0.1 M HClO₄ electrolyte (70 % Suprapur - Merck) using the anodic scan between 0.05 and 1.2 V vs. RHE, with a scan rate of 20 mV/s and rotation rate of 1600 rpm at room temperature. All scans were normalised for electrochemically active surface area and corrected for ohmic drop measured by impedance spectroscopy and capacitive current in an oxygen free electrolyte. Error margins were obtained from 2 to 4 repeats for each data point.

Further minor changes to the manuscript

Removed references: Unfortunately, we had to remove some references after including additional references in response to above points to adhere with the journal limit of 70 citations. We mostly removed references from the introduction, with preference given to situations where more than one citation was used to point to similar literature.

Moved figures: We moved the EXAFS data into the supplemental to adhere with Journal guidelines regarding display items (10 items max.).

REVIEWERS' COMMENTS:

Reviewer #1 (Remarks to the Author):

Dear authors

Thank you for addressing all my comments in your rebuttal. However, if i may for future publications it would be better if you would avoid the use of Nafion during RDE measurements, as it strongly affects the mass activity due to sulfonic poisoning of the Pt particles. Thus by avoiding Nafion in RDE a more comprehensive and less confusing understanding of the data can be made, especially when arguments of enhanced mass activity are make due to the change of the support.

Reviewer #2 (Remarks to the Author):

The authors have responded satisfactorily to my previous three questions. The evidence for the direction of charge transfer is convincing. The claim of strong-metal-support interaction between Pt nanoparticle and the support can be justified which explains the small-size and stability of the Pt nanoparticle on the support.

The paper can now be published.

Reviewer #3 (Remarks to the Author):

The authors have addressed all my comments and concerns. The manuscript have improved significantly form the first version and I believe that the contributions are original and relevant in the field. I recommend its publication in Nature Communications.

Rebuttal Letter – Revision 2

We thank all reviewers for their help preparing the manuscript for publication. A point-by-point response to the comments on revision 2 is given below.

Response to Reviewer 1

Revision 2: R1(1) Thank you for addressing all my comments in your rebuttal. However, if I may, for future publications it would be better if you would avoid the use of Nafion during RDE measurements, as it strongly affects the mass activity due to sulfonic poisoning of the Pt particles. Thus, by avoiding Nafion in RDE, a more comprehensive and less confusing understanding of the data can be made, especially when arguments of enhanced mass activity are made due to the change of the support.

We thank Reviewer 1 for the insightful discussion and agree that ionomer-free electrodes have appeal for future fundamental studies as they simplify the system.

We would like to take the opportunity to further comment on the interpretation of XPS spectra that Reviewer 1 commented on before: Revision1: R1 (3) At the XPS data, the O spectra are missing, rising questions, especially due to the presence of a clear PtO shoulder on the Pt spectra for the case of 10 eq. wt% Pt/B4C. It is widely known that the binding energy of PtO shifts to higher energies compared to metallic Pt. The fitting of the Pt spectra would have cleared this open question- if provided, as it is essential towards the claim of metal support interaction due to the observed shift in the binding energy.

The core level shifts of supported metallic nanoparticles have three contributions: (i) a shift to higher (lower) binding energy due to band filling (depletion) in the valence band upon electron transfer to (from) the nanoparticle, (ii) a similar shift in support core levels for semi-conducting or insulating supports, and (iii) a shift to higher binding energies due to final state effects.

(i) Band filling of catalyst particles. The shift of nanoparticle core levels due to electron transfer into the Pt valence d-band has been extensively discussed by Watanabe and co-workers. Figures 4 and 7 in [J. Phys. Chem B (2006) 23489] are of particular interest. They show a shift to higher binding energies of dealloyed Pt alloy catalysts (Pt-rich shell of about 2-3nm; estimated from a reduction of the Co XPS intensity; see paper for detail). These shifts are rationalized by an electron transfer from the Co-rich alloy to the Pt shell as schematically shown in Figure 4 of [J. Phys. Chem B (2006) 23489]. It is important to note that the Pt shell is approximately of the same thickness as the size of our nanoparticles. There is, therefore, a close analogy between the electron donating core in the studies of Watanabe et al. and the electron donating support in our study, which further corroborates the analogy to Pt alloys that was already mentioned in the manuscript when discussing the XANES results.

Theoretical work by Henkelman and co-workers supports that view [J. Chem. Phys. 130 (2009) 194504]. They used DFT to investigate core-shell nanoparticles (Pd shell). Figure 8 in their paper is of particular interest. It shows the shift in Pd d-band centre (relative to the Fermi level) as a function of donated electrons from the core into the shell of core-shell particles

(Pd shell). It has recently been experimentally demonstrated that electron transfer between support and Pt nanoparticles can reach values of the order of 0.1 e/at [Nature Material 15 (2016) 1-6]. For Pd, this corresponds to a shift of about 0.4 eV to higher binding energies in Henkelman's model, in good qualitative agreement with the observed order of magnitude of core level shifts although it should be mentioned that Henkelman investigated monolayer shells.

(ii) Band filling of support surface. In principle, a similar band filling argument should be made for the support side. There is, however, usually much more support material than catalyst, providing a much larger density of states. One would, therefore, not generally expect a pronounced band filling (depletion) effect if there is sufficient density of states near the Fermi level. However, the occupied valence density of states near the Fermi level can be small in n-type semiconductors. Support-side core level shifts have been demonstrated for a gold decorated SrTiO₃ surface [Surf. Sci. (2010) 548-554]. The system is close to our system in the sense that the support work function is much smaller than the particle work function (Au ~5.4eV; support ~3.5eV) and electrons are, therefore, transferred into Au. This is accompanied by a pronounced shift of support core levels to lower binding energies by up to 0.3 eV, demonstrating that support side core level shifts should be expected for semi-conducting samples. As a side note, SrTiO₃ exhibits a very low flat band potential (equivalent of pzc for semi-conductor electrodes) [B. Jap. Chem. Soc. (1976) 355-358] underlining the connection between work function and pzc mentioned in the manuscript.

The figure to the right shows the valence band spectrum of our BC supports as obtained from XPS. It compares the region close to the Fermi level between uncatylsed BC and BC catalyzed with 10 eq. wt% Pt. There is only very small density of states up to about 2 eV below the Fermi level for uncatylsed BC demonstrating that the material exhibits electronic properties of a n-type semiconductor with a band gap of somewhat below 2 eV in reasonable agreement with bulk B₄C (band gap 2.1 eV). In contrast, the catalyzed sample shows substantial density of states right at the Fermi level, which we believe to be mostly Pt d-band states.

This indicates that one generally should expect support side core level shifts to contribute to the relative shift between Pt 4f and C 1s core levels for BC. It is, however, important to realize that particle and support core level shifts contribute to the difference in an additive manner. If particle levels shift to higher binding energy due to charge transfer into the particle, support levels will shift to lower binding energy, and both contribute in the same manner to the reduction of the difference between them (Pt 4f levels are higher than the C1s levels).

(iii) Final state effects. It has long been discussed that long-lived holes can contribute to the binding energies of metallic clusters and nanoparticles obtained from XPS on supports with limited conductivity. Because of the limited conductivity of supports, the hole left behind in the nanoparticle upon photo excitation is argued to be long-lived on the timescale of XPS experiments. The hole generates a positive Coulomb field, stabilizing the remaining electrons. This additional energy contribution is observed as a positive contribution to the binding energy in XPS (including a shift of states at the Fermi level). The effect has been quantified for Au clusters on vitreous carbon [PRL 51 (1983) 2310-2313] from a shift of the apparent Fermi level as a function of Au loading. Our Pt loadings correspond to about $0.4 - 1.5 \cdot 10^{15}$ atoms per cm^2 . Although comparison between coverage and particle size is limited without further knowledge of the respective metal growth, the effect for Au on vitreous carbon for similar loadings is of the order of 0.5 eV at most. If one accepts the density of states on both sides of the interface as an indicator for electron transfer propensity, the effect might be particularly pronounced for Au owing to the relatively low density of states at the Fermi level (which is dominated by a rather wide s band; the first Au d-band is located about 2eV below the Fermi level [PRB 23 (1981) 505-508]). As we have shown above, the valence band spectrum for catalyzed BC clearly shows occupancy at zero binding energy, indicating that electrons at the Fermi level are not significantly shifted to higher binding energies even at the lowest loading. This might be taken as an indication that final state effects are less relevant in our case.

We conclude that the observed relative shift of Pt 4f and C 1s levels is in agreement with a model considering electron transfer from support to particle with a possible minor contribution from final state effects.

Response to Reviewer 2

Revision 2: R2(1) The authors have responded satisfactorily to my previous three questions. The evidence for the direction of charge transfer is convincing. The claim of strong-metal-support interaction between Pt nanoparticle and the support can be justified which explains the small-size and stability of the Pt nanoparticle on the support. The paper can now be published.

We thank Reviewer 2 for the productive discussion and support of our manuscript.

Response to Reviewer 3

Revision 2: R3(1) The authors have addressed all my comments and concerns. The manuscript has improved significantly from the first version and I believe that the contributions are original and relevant in the field. I recommend its publication in Nature Communications.

We thank Reviewer 3 for the supportive comments and insightful critique of our manuscript.